# CTRP9 engages AdipoR1 and promotes T cell glycolysis and immunity

Kunming Li [ID][1,4], Jiansong Zhang[1,2,4], Kang Li[1,2,4], Haokai Chen[1,2,4], Wenhai Deng[3,4], Wenzhuo Rao[1], Ming Geng[1], Yuying Zheng[1], Xiumei Wei [ID][1,2] & Jialong Yang [ID][1,2 ✉]

## Abstract

**The adiponectin (ADPN) receptor (AdipoR) modulates T-cell responses, but its effects remain controversial since signaling can either promote or inhibit T-cell function. Interaction with the ligand ADPN inhibits T-cell responses, but given the existence of multiple AdipoR ligands, we hypothesize that ligand diversity underlies its differential effect in T-cell immunity. To test this, we use tilapia and mouse models. Tilapia encodes AdipoR1 but lacks ADPN. Instead, an alternative adipokine, CTRP9, engages AdipoR1. We find CTRP9–AdipoR1 interaction triggers $Ca^{2+}$ influx and activates the CaM–CaMKKβ–AMPK pathway, facilitating crosstalk with TCR signaling. This cascade enhances T-cell activation, proliferation, and antimicrobial immunity by promoting glycolysis. In mice, CTRP9 similarly enhances T-cell activation, proliferation, and cytokine production and improves the efficacy of anti-CD19 CAR-T cells in eliminating B-cell lymphoma in vitro. These findings reveal an evolutionarily conserved role of CTRP9 in promoting T-cell immunity, in contrast to the inhibitory effect exerted by ADPN. Mechanistically, CTRP9 and ADPN exert distinct effects on T-cell metabolism; CTRP9 enhances T-cell glycolysis, whereas ADPN suppresses it. We therefore propose ligand selectivity as a determinant of AdipoR1-dependent T-cell immune outcomes.**

**Keywords** AdipoR1; CTRP9; Adiponectin; T Cell Immunity; Glycolysis
**Subject Categories** Immunology; Metabolism; Signal Transduction

## Introduction

Metabolism plays a crucial role not only in energy production and cell growth but also in immune regulation. Recent studies have highlighted the importance of metabolic pathways and their reprogramming in shaping T-cell function and fate (Franco et al, 2020; Lim et al, 2022). The molecular foundation of immunometabolism involves a complex interplay between metabolic receptors, transporters, signaling molecules, and transcription factors with T-cell signaling pathways (Cai et al, 2024;

Chapman et al, 2020; Klein Geltink et al, 2020). Adiponectin receptors (AdipoRs) are seven-transmembrane proteins widely expressed in muscle cells and adipocytes (Tanabe et al, 2015). Unlike classical G-protein-coupled receptors (GPCRs), AdipoRs possess an intracellular N-terminus and an extracellular C-terminus and cannot interact with G proteins (Deckert et al, 2006; Yamauchi et al, 2003), placing them in a distinct class of seven-transmembrane receptors (Villa et al, 2009). To date, three AdipoRs—AdipoR1, AdipoR2, and T-cadherin—have been identified in mammals (Wang et al, 2014), where they activate downstream signaling pathways, including AMP-activated protein kinase (AMPK) and peroxisome proliferator-activated receptors (PPARs), to regulate glucose and lipid metabolism (Kadowaki et al, 2006; Iwabu et al, 2010; Matsuzawa, 2010; Shetty et al, 2009). Dysregulation of AdipoR signaling contributes to metabolic disorders such as metabolic syndrome, diabetes, cardiovascular disease, and insulin resistance (Kadowaki et al, 2006; Iwabu et al, 2010; Matsuzawa, 2010; Shetty et al, 2009). In mice, AdipoR1 deficiency significantly reduces AMPK activity and promotes gluconeogenesis, leading to elevated blood glucose levels (Yamauchi et al, 2002; Yamauchi et al, 2007). Notably, AdipoRs have also been found to be expressed in T cells, where they actively influence cellular function and fate. For example, AdipoR1 promotes IL-10 secretion and inhibits T-cell activation by suppressing NF-κB signaling in dendritic cells (Tan et al, 2014). Moreover, AdipoR signaling reduces the production of IFN-γ and TNF-α by T cells, thereby reducing the self-renewal capacity of hematopoietic stem cells (Meacham et al, 2022). Additionally, AdipoR1 inhibits Th1 and Th17 cell differentiation through antigen presentation-related pathways (Xiao et al, 2017). These findings suggest that AdipoRs negatively regulate T-cell immunity. However, this idea remains controversial; some other studies have shown that AdipoR1 deficiency suppresses T-cell-related gene expression, reduces HIF-1α-mediated glycolysis, and impairs Th17 cell differentiation (Zhang et al, 2020). Together, these findings suggest that AdipoRs have diverse, context-dependent effects on T-cell immunity; however, the factors underlying these differing outcomes remain unclear.

Adiponectin (ADPN), the well-characterized ligand of AdipoRs, is an adipokine primarily secreted by adipocytes and plays a crucial role in regulating glucose and lipid metabolism, improving insulin resistance, and modulating anti-inflammatory and antioxidant responses (Li et al, 2010). Recent studies have reported that ADPN negatively regulates T-cell immunity at multiple levels. For example, ADPN reduces T-cell responsiveness by inducing apoptosis, inhibiting proliferation, and suppressing cytokine

[1]State Key Laboratory of Estuarine and Coastal Research, School of Life Sciences, East China Normal University, 200241 Shanghai, China. [2]Laboratory for Marine Biology and Biotechnology, Qingdao Marine Science and Technology Center, 266237 Qingdao, China. [3]School of Laboratory Medicine and Life Science, Wenzhou Medical University, 325035 Wenzhou, China. [4]These authors contributed equally: Kunming Li, Jiansong Zhang, Kang Li, Haokai Chen, Wenhai Deng. ✉E-mail: jlyang@bio.ecnu.edu.cn

production (Wilk et al, 2011). Additionally, ADPN promotes Treg expansion and IL-10 secretion while inhibiting Th17 cell differentiation and function (Li et al, 2019; Ramos-Ramírez et al, 2021). These effects are consistent with findings suggesting that AdipoR signaling suppresses T-cell immunity. Notably, ADPN is not the only ligand for AdipoRs. CTRP9, another adipokine with collagen and globular domains similar to those of ADPN, also binds and activates AdipoRs, regulating glucose and lipid metabolism (Guan et al, 2022). However, the effect of CTRP9 on T-cell responses remains unknown. Whether AdipoRs mediate the multidirectional regulation of T-cell immunity by binding different ligands is a question worth exploring. Moreover, AdipoR1 is an evolutionarily conserved receptor, but it remains unclear whether AdipoR1-mediated regulation of T-cell immunity is an independently acquired feature in mammals or a gradually evolved function common to vertebrates. Furthermore, whether and how CTRP9 and ADPN have together shaped T-cell immunity during evolution is an intriguing question.

From an evolutionary viewpoint, fish represent the lowest extant vertebrates possessing T cells. Although AdipoRs are widely expressed in fish species such as zebrafish, large yellow croaker, and black carp (Ji et al, 2020; Rastegar et al, 2019; Wu et al, 2018), their role in T-cell immunity remains unexplored. Our previous studies in Nile tilapia (*Oreochromis niloticus*) have elucidated the mechanisms underpinning Th1 cell differentiation and T-cell homeostasis (Ai et al, 2022; Li et al, 2023a; Zhang et al, 2024c; Zhang et al, 2023) and demonstrated that fish regulate T-cell responses through metabolic pathways such as glutaminolysis and glycolysis (Li et al, 2023b; Wei et al, 2019; Wei et al, 2020b). These findings indicate a crosstalk between metabolism and immune signaling in fish T cells. However, whether and how AdipoR1 signaling regulates T-cell immunity in fish, as well as which ligand activates this receptor, remain unknown.

In this study, we used tilapia and mouse models to evaluate the differential regulation of CTRP9 and ADPN on T-cell immunity. Tilapia encodes AdipoR1 and CTRP9 but has lost ADPN during evolution, making it an ideal model to investigate the role of the CTRP9–AdipoR axis in T-cell immunity without interference from ADPN. Using this model, we suggest that tilapia CTRP9 binds to AdipoR1 and activates the $Ca^{2+}$–CaMKKβ–AMPK signaling, which promotes T-cell immunity by regulating glycolysis. In mouse, CTRP9 enhances the activation and proliferation of $CD4^+$ and $CD8^+$ T cells and increases their cytokine production, representing a previously unknown function of CTRP9. Moreover, CTRP9 boosts the activity of anti-CD19 CAR-T cells against B-cell lymphoma. Notably, we propose that two distinct adipokines—ADPN and CTRP9—elicit opposing effects on T-cell glycolysis in mouse, potentially explaining the seemingly contradictory observations regarding differential regulation of T cells by AdipoR.

## Results

### Tilapia lacks ADPN but retains CTRP9 to bind AdipoR1

We identified an *AdipoR1* gene in the cold-blooded vertebrate Nile tilapia. Tilapia *AdipoR1* encodes a peptide comprising 376 amino acids, with a molecular weight of 42.5 kDa. This AdipoR1 is a seven-transmembrane protein with four extracellular segments: ECL1, ECL2, ECL3, and CTR (Appendix Fig. S1A). The amino acid

sequence of tilapia AdipoR1 shares high similarity with its homologs in other fish species, amphibians, birds, and mammals (Appendix Fig. S1A). Notably, the arrangement of the functional domains of AdipoR1 in tilapia is similar to that in mouse: a RING domain with E3 ubiquitin-protein ligase activity followed by a HlyIII domain characteristic of seven-transmembrane proteins (Appendix Fig. S1B). In addition, tilapia and mouse AdipoR1 exhibit a high degree of similarity in their tertiary structures (Fig. 1A). Phylogenetic analysis revealed that tilapia AdipoR1 was clustered with its homologs from other teleost species (Fig. 1B). Although not well conserved across species, the ligand of AdipoRs, ADPN, is widely present in vertebrates (Appendix Fig. S1C). However, a search of the NCBI and Ensembl genome databases of Nile tilapia using ADPN amino acid sequences from various fish species returned no matches, suggesting that Nile tilapia lacks ADPN. Remarkably, the loss of ADPN was observed throughout the genus Tilapia and may even be prevalent in the family Cichlidae, despite the presence of two AdipoRs in these species (Fig. 1C). In contrast, non-cichlid species possess 1-6 copies of ADPNs or AdipoRs, suggesting a gene expansion event in some fish species (Fig. 1C). The presence or absence of ADPNa/ADPNb in fish was further confirmed by a collinearity analysis of the gene locus (Fig. 1D,E). These observations collectively suggest that tilapia possesses AdipoRs but has lost its ligand ADPN.

Next, we sought to identify the ligand that binds AdipoR in tilapia. A deep search in the tilapia genomic database identified a homologous adipokine of *ADPN*: *CTRP9*. Tilapia CTRP9 shares moderate similarities in amino acid sequences, functional domains, and tertiary structures with CTRP9 in other vertebrates (Fig. 1F; Appendix Fig. S2A,B). As predicted, tilapia CTRP9 can closely interact with AdipoR1 through hydrogen bonds and salt bridges (Fig. 1G; Appendix Fig. S2C). To further confirm this interaction, we prepared a His-tagged recombinant tilapia CTRP9 (rOnCTRP9) (Appendix Fig. S2D) and a rat anti-tilapia AdipoR1 polyclonal antibody, which can specifically recognize the immunogen (Appendix Fig. S2E) and tilapia AdipoR1 in spleen leukocytes (Fig. 1H), and identify $AdipoR1^+$ lymphocytes (Fig. 1I). The tilapia lymphocytes were incubated with rOnCTRP9 and stained with anti-His and anti-AdipoR1 antibodies. Flow cytometry revealed that $AdipoR1^+$ lymphocytes, but not $AdipoR1^-$ lymphocytes, can bind rOnCTRP9 (Fig. 1J). Thus, these findings demonstrate that the cold-blooded vertebrate Nile tilapia lacks ADPN but has retained CTRP9 to bind AdipoR1.

### AdipoR1 is associated with T cells and participates in the antibacterial immune response

Next, we investigated whether AdipoR1 is involved in the T-cell-mediated immune response of tilapia. In tilapia infected by *Aeromonas hydrophila*, *AdipoR1* transcription in spleen lymphocytes was upregulated at 5 days post infection (dpi) (Fig. 2A). Consistent with this finding, the protein level of AdipoR1 was elevated at 4 and 7 dpi (Fig. 2B), indicating an involvement of AdipoR1 in the antibacterial primary immune response. In uninfected tilapia, approximately 25% of spleen lymphocytes expressed AdipoR1, and the proportion of these was markedly higher at 5 dpi (Fig. 2C,D). The expansion of $AdipoR1^+$ lymphocytes during *A. hydrophila* infection was further confirmed by the immunofluorescence assay (Fig. 2E). Next, we addressed the association between AdipoR1 and T-cell lineage in tilapia. Except for a

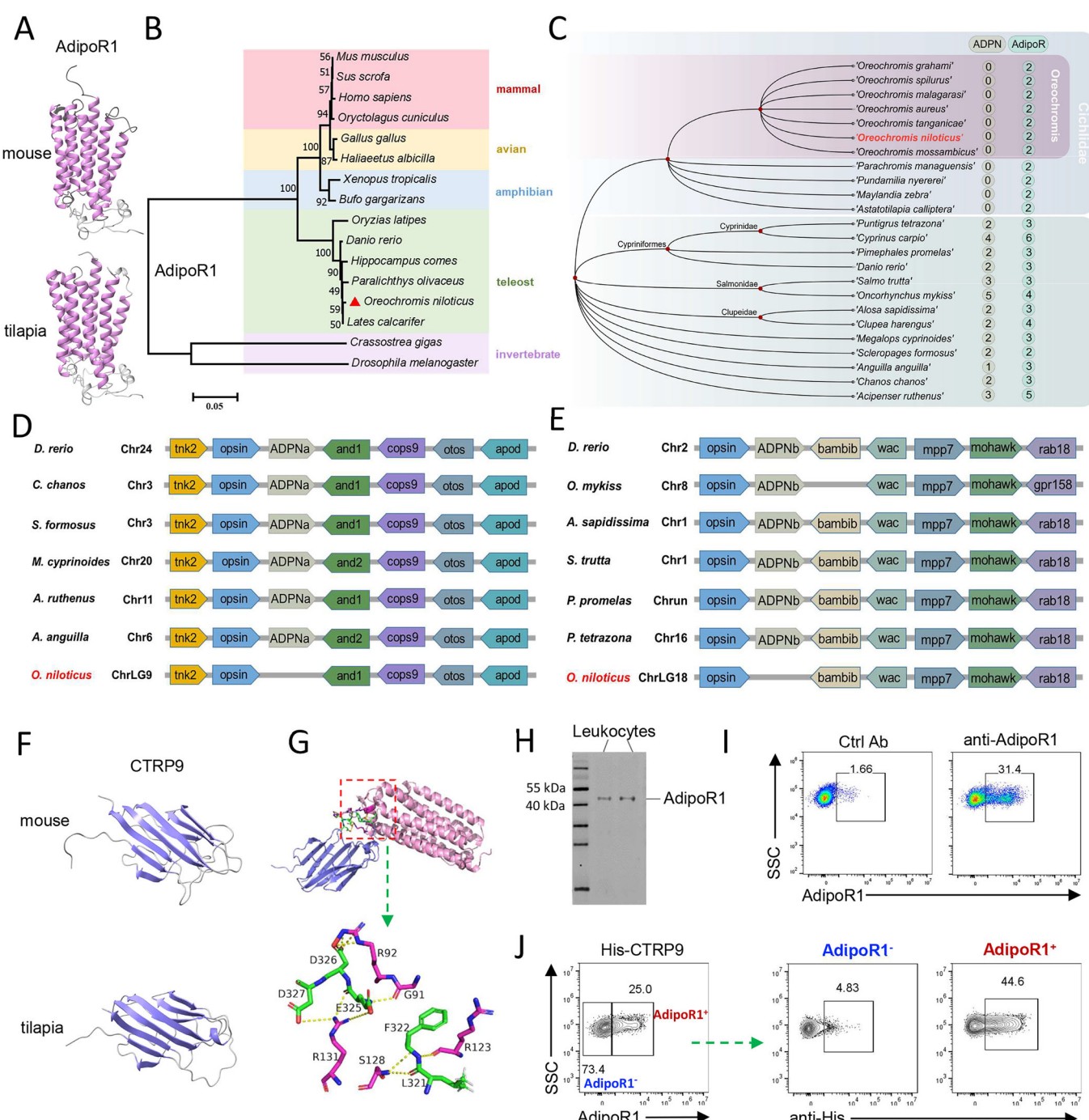

**Figure 1. Nile tilapia has lost the ADPN, but retains AdipoR1 and CTRP9.**

(A) Prediction of the tertiary structure of AdipoR1 from mouse and tilapia. (B) A phylogenetic tree for AdipoR1 was constructed with the amino acid sequences of AdipoR1 from the indicated species. The tree was constructed in MEGA7 by using the neighbor-joining (NJ) method with 1000 bootstrap replications. (C) Copy numbers of ADPN and its receptor genes in tilapias and other fish species. (D, E) Collinearity analysis of ADPNa (D) and ADPNb (E) based on genomics of the Nile tilapia and other fish species. (F) Prediction of the tertiary structure of CTRP9 from mouse and tilapia. (G) Global docking results for the CTRP9–AdipoR1 complex obtained using the ZDOCK server. Green sticks indicate salt bridges, and blue and pink sticks indicate CTRP-9 and AdipoR1, respectively. (H, I) Identification of tilapia AdipoR1 antibodies. Western blot using tilapia leukocytes showed the specificity of AdipoR1 antibody (H). Flow cytometry showing AdipoR1 staining in gated lymphocytes of tilapia (I). (J) Spleen leukocytes were incubated with recombinant His-tag CTRP9 and then stained with anti-His and anti-AdipoR1 antibody. Flow cytometry showing the frequency of CTRP9-binding cells among AdipoR1+ or AdipoR1- lymphocytes. Data information: Experiments in (H–J) were repeated three times.

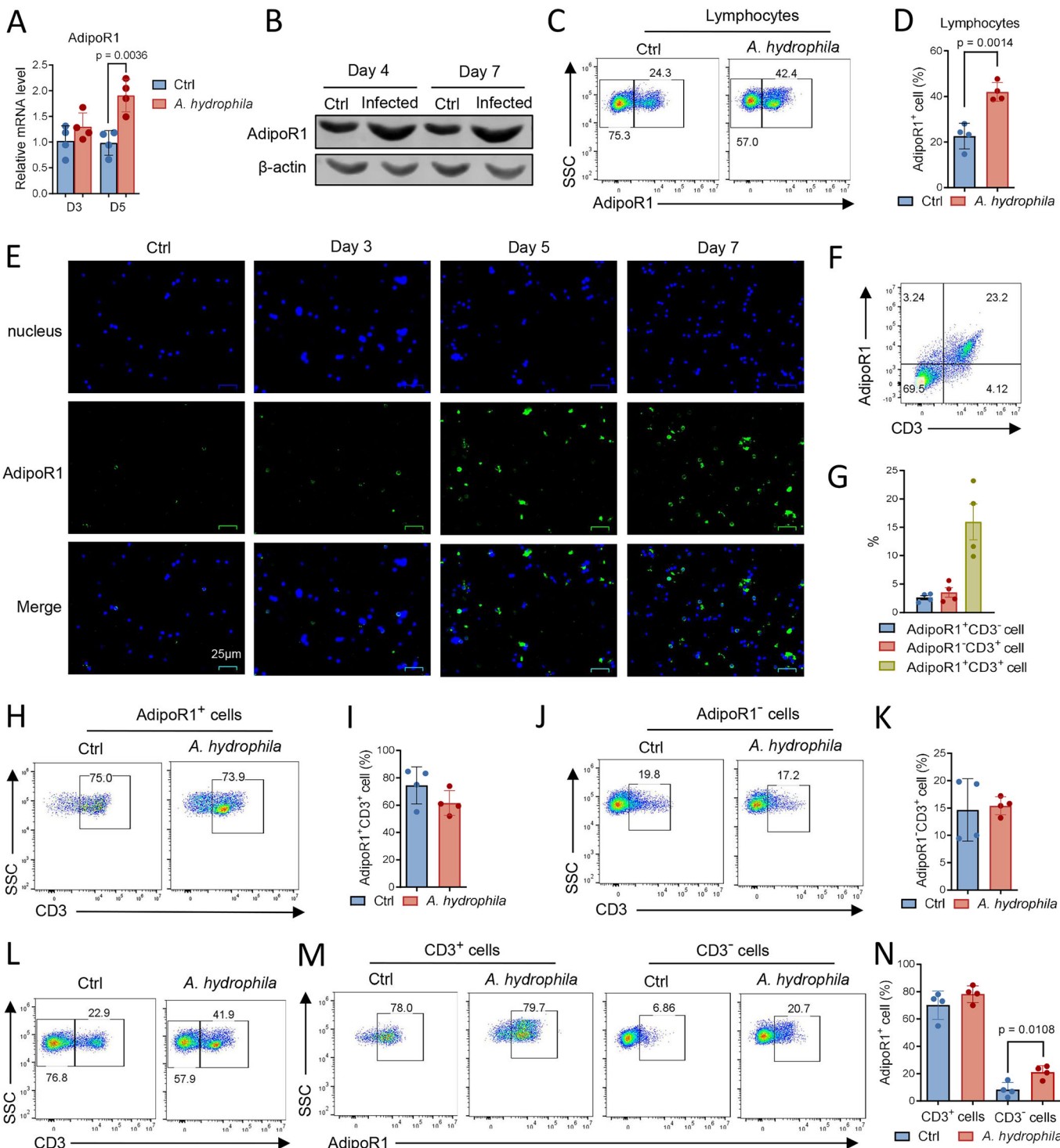

**Figure 2. AdipoR1 is involved in the T-cell immune response of Nile tilapia.**

Tilapia individuals were infected with *A. hydrophila*, and spleen leukocytes were isolated at the indicated times. (**A**) Relative mRNA levels of AdipoR1 by qPCR on 3 and 5 dpi, n = 4. (**B**) Protein levels of AdipoR1 by Western blot on 4 and 7 dpi. (**C, D**) Representative FACS plots (**C**) and bar figure (**D**) showing the AdipoR1+ cells in gated lymphocytes, n = 4. (**E**) Immunofluorescence analysis showing the AdipoR1+ cells at the indicated dpi. (**F, G**) Representative FACS plots (**F**) and bar figure (**G**) showing the CD3 and AdipoR1 staining in gated lymphocytes from uninfected individuals, n = 4. (**H–K**) Representative FACS plots (**H, J**) and bar figures (**I, K**) showing the CD3 staining in gated AdipoR1+ or AdipoR1- cells on 5 dpi, n = 4. (**L**) Representative FACS plots showing the CD3 staining in gated lymphocytes on 5 dpi. (**M, N**) Representative FACS plots (**M**) and bar figures (**N**) showing AdipoR1 staining in gated CD3+ or CD3- cells on 5 dpi, n = 4. Data information: n stands for biological replicates. Error bars indicate mean ± SEM. Significance between the groups was determined by a two-tailed Student's t test.

small proportion of CD3$^+$ T cells not expressing AdipoR1 or AdipoR1$^+$ cells not expressing CD3, spleen T cells co-expressed these two receptors (Fig. 2F,G). Although bacterial infection induced the expansion of AdipoR1$^+$ cells (Fig. 2C,D), it did not alter the T-cell composition in the AdipoR1$^+$ or AdipoR1$^-$ cell population (Fig. 2H–K). It was observed that in both control and *A. hydrophila*-infected tilapia, CD3$^+$ T cells made up the majority of the AdipoR1$^+$ cell population (Fig. 2H,I), whereas non-T cells contributed only a small fraction (Fig. 2J,K). As reported previously (Li et al, 2023c; Wei et al, 2020b), *A. hydrophila* infection caused a significant expansion of CD3$^+$ T cells in the spleen (Fig. 2L). The frequencies of AdipoR1$^+$ cells within the T-cell population were high and comparable between control and infected tilapia (Fig. 2M,N). In contrast, only a small fraction of non-T cells was AdipoR1-positive, but the frequency of this subset increased following bacterial infection (Fig. 2M,N). Together, these findings suggest that AdipoR1 is strongly associated with T cells and plays a role in the antibacterial T-cell immune response in tilapia.

## AdipoR1 signaling is essential for T-cell immunity

Because AdipoR1 was found to be associated with T cells, we further elucidated its role in the T-cell immune response of tilapia. The AdipoR1 antibody was used to block AdipoR1 signaling during bacterial infection. *A. hydrophila* infection induced a robust T-cell response in tilapia at 5 dpi, as evidenced by increased spleen lymphocytes (Appendix Fig. S3A) and T-cell percentages and absolute numbers (Fig. 3A,B; Appendix Fig. S3B). However, blocking AdipoR1 severely impaired the expansion of lymphocytes (Appendix Fig. S3A) and T cells (Fig. 3A,B; Appendix Fig. S3B). Moreover, the AdipoR1 antibody significantly impaired the *A. hydrophila*-induced IL-2 production and CD122 (IL-2Rβ) expression in gated spleen T cells (Appendix Fig. 3C–E), indicating the crucial role of AdipoR1 in regulating T-cell activation. Further investigation suggested that the reduced T-cell expansion upon AdipoR1 blockade was due to impaired T-cell proliferation (Fig. 3F,G) and increased T-cell apoptosis (Fig. 3H,I), suggesting that AdipoR1 signaling is also indispensable for the proliferation and survival of tilapia T cells. We then sought to determine whether AdipoR1 signaling facilitates the ability of tilapia T cells to eliminate infection. Administration of the AdipoR1 antibody during bacterial infection impaired the inducible expression of cytotoxic genes *Perforin A* and *Granzyme B* (Appendix Fig. S3C) and pro-inflammatory cytokines *IFN-γ*, *TNF-α*, and *IL-6* (Appendix Fig. S3D) and reduced the ability of T cells to produce Granzyme B (Fig. 3J,K). These defects in effector function hindered the ability of T cells to eliminate bacterial infection (Fig. 3L), consequently leading to higher mortality (Fig. 3M). Altogether, our findings suggest that AdipoR1 signaling is indispensable for T-cell activation, proliferation, and effector function in tilapia.

## CTRP9 and AdipoR1 activate AMPK signaling in T cells via the Ca$^{2+}$–CaMKKβ axis

Next, we investigated the precise signaling pathways downstream of AdipoR1 in tilapia T cells. AdipoR1 is strongly associated with Ca$^{2+}$ signaling (Iwabu et al, 2010). Our previous studies have shown that TCR signaling induces Ca$^{2+}$ influx and CaM expression in tilapia (Wei et al, 2020a). In the present study, we identified a CaMKKβ gene in Nile tilapia. The amino acid sequence of tilapia CaMKKβ,

particularly the S_TKc domain responsible for the catalytic activity, is evolutionarily conserved among vertebrates (Appendix Fig. S4A). Notably, the functional domain and tertiary structure of tilapia CaMKKβ showed high similarity to its mouse homolog (Appendix Fig. S4B,C). In addition, tilapia CaMKKβ was closely clustered with its counterparts from other teleosts in the phylogenetic tree (Appendix Fig. S4D). To determine whether tilapia AdipoR1 induces Ca$^{2+}$ influx, we used the agonist AdipoRon to activate AdipoR1 in tilapia spleen leukocytes. Upon agonist administration, a rapid and sustained Ca$^{2+}$ influx was observed in the lymphocyte population (Fig. 4A; Appendix Fig. S5A), which subsequently enhanced the expression of CaM and CaMKKβ at both mRNA and protein levels (Fig. 4B; Appendix Fig. S5B). AdipoR1 activation was correlated with increased AMPK transcription and enhanced AMPK phosphorylation (Fig. 4B; Appendix Fig. S5B). Specifically, the increased CaM and CaMKKβ expression and AMPK phosphorylation upon AdipoRon administration were observed in gated CD3$^+$ T cells (Fig. 4C; Appendix Fig. S5C). To confirm whether AdipoR1 activates AMPK via CaMKKβ, we used STO-609 to inhibit CaMKKβ activity. Inhibition of CaMKKβ significantly impaired AdipoR1 activation-induced AMPK phosphorylation (Fig. 4D), indicating that CaMKKβ is essential for AdipoR1-mediated AMPK activation in tilapia. Next, we determined whether AdipoR signaling is initiated by its ligand. Both in vitro and in vivo administration of rOnCTRP9 significantly induced the expression of CaM and CaMKKβ in spleen leukocytes at mRNA and protein levels (Fig. 4E; Appendix Fig. S5D–F), and elevated the transcription and phosphorylation of AMPK (Fig. 4E; Appendix Fig. S5D–F). To exclude potential influences from non-T-cell lineages, we sorted CD3$^+$ T cells from tilapia and treated them with rOnCTRP9 in vitro. rOnCTRP9 administration activated the Ca$^{2+}$–CaMKKβ–AMPK signaling axis in purified T cells as well (Appendix Fig. S5G), suggesting its direct regulation of this signaling in T cells. Thus, our findings suggest that AdipoR1 senses the signals from CTRP9 to activate the Ca$^{2+}$–CaMKKβ–AMPK axis in tilapia T cells.

## CTRP9 initiates the AdipoR1–Ca$^{2+}$–CaMKKβ axis in T cells to regulate antibacterial response

Given that AdipoR1 signaling activates the Ca$^{2+}$–CaMKKβ–AMPK pathway in tilapia T cells, we further investigated the role of this pathway in T-cell immunity. The Ca$^{2+}$–CaMKKβ–AMPK pathway was detected to be involved in the antibacterial immune response after *A. hydrophila* infection (Appendix Fig. S6A,B). As described previously (Li et al, 2023c), PHA stimulation enhanced the phosphorylation of NF-κB and ERK1/2 (Appendix Fig. S6C), indicating robust T-cell activation. However, inhibition of CaMKKβ lead to defects in T-cell activation (Appendix Fig. S6C,D). Similarly, during bacterial infection, CaMKKβ inhibition impaired the expansion of lymphocytes (Appendix Fig. S6E) and T cells (Appendix Fig. S6F–H), reduced the proportion of T cells producing IL-2 (Appendix Fig. S7A,B) or expressing CD122 (Appendix Fig. S7C), and exacerbated *A. hydrophila*-induced T-cell apoptosis (Appendix Fig. S7D,E). Notably, the loss of CaMKKβ activity inhibited the ability of tilapia leukocytes to express *Perforin A*, *Granzyme B*, *IFN-γ*, *TNF-α*, and *IL-6* (Appendix Fig. S7F) and reduced the frequency of Granzyme B-producing T cells (Appendix Fig. S7G,H), consequently impairing the ability

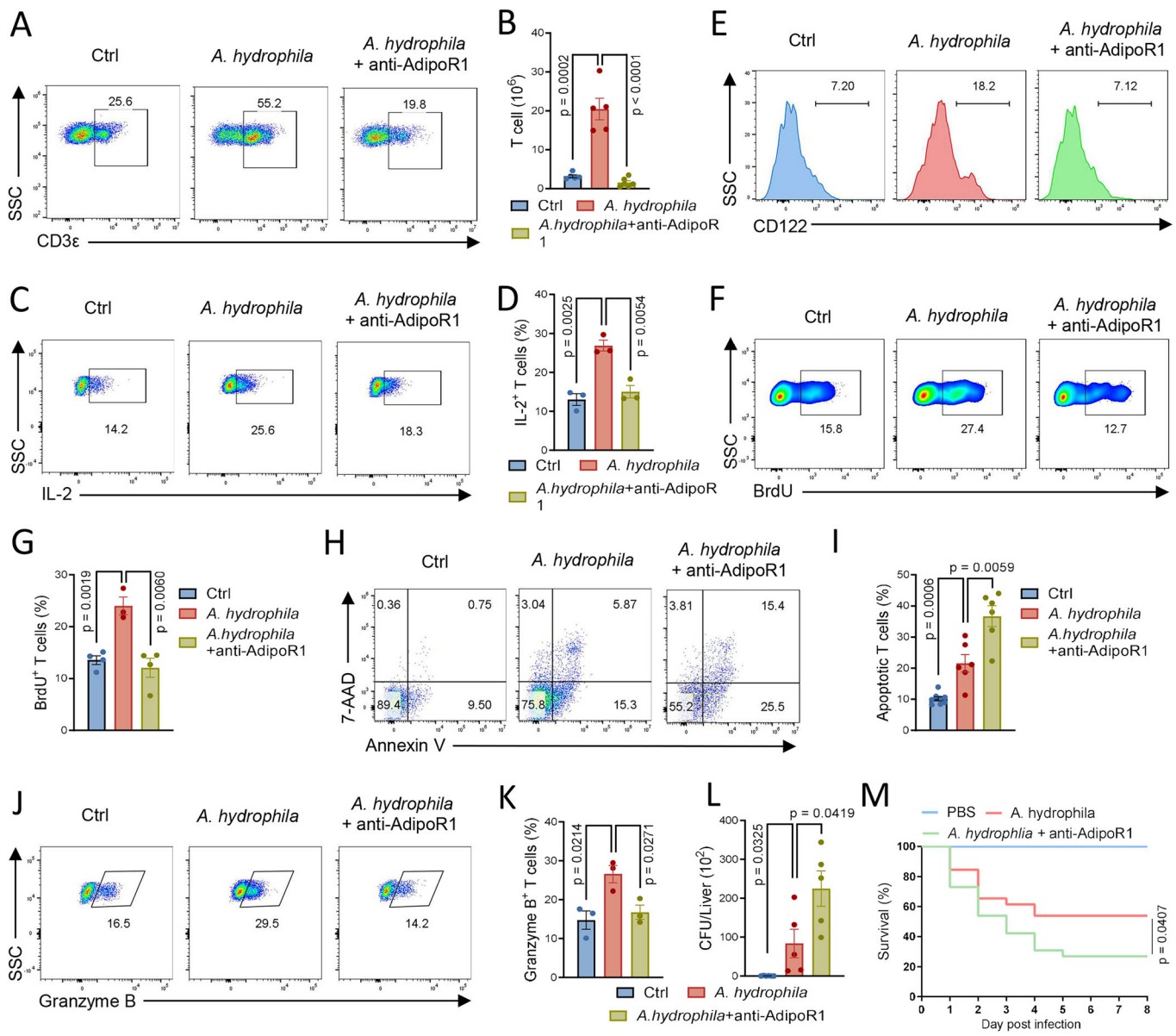

**Figure 3. AdipoR1 is crucial for T-cell immunity in tilapia.**

Tilapia individuals infected with *A. hydrophila* were *i.p.* injected with purified AdipoR1 antibody on days 2, 3, 4, and 5, and the animals were sacrificed at the indicated time points for assay. (A–K) Spleen leukocytes were isolated on 5 dpi. Representative FACS plots (A) and absolute number (B) of CD3⁺ T cells were shown, *n* = 4. Tilapia individuals were *i.p.* injected with BFA 6 h before sacrifice. Representative FACS plots (C) and bar figure (D) showing the percentage of IL-2⁺ T cells in gated CD3⁺ T cells, *n* = 3. Histograms showing the levels of CD122 in gated CD3⁺ T cells (E). Tilapia individuals were *i.p.* injected with BrdU 1 day before sacrifice, and spleen lymphocytes were isolated for assays, and representative FACS plots (F) and bar figure (G) showing the percentage of BrdU⁺ T cells in gated CD3⁺ T cells, *n* = 3–4. Representative FACS plots (H) and bar figure (I) showing the 7-AAD and Annexin V staining on gated CD3⁺ T-cell population, *n* = 5. Tilapia individuals were *i.p.* injected with BFA 6 h before sacrifice, and spleen lymphocytes were isolated for assays. Representative FACS plots (J) and bar figure (K) showing the percentage of granzyme B⁺ T cells in gated CD3⁺ T cells, *n* = 3. (L) *A. hydrophila* titers in the liver of infected tilapia on 6 dpi, *n* = 5–6. (M) Kaplan–Meier survival plot showing the survival percentage of tilapia, *n* = 26. Data information: *n* stands for biological replicates. Error bars indicate mean ± SEM. Significance between the groups was determined by a two-tailed Student's *t* test.

to control bacterial infection (Appendix Fig. S7I). Therefore, our results suggest that the AdipoR1–CaMKKβ pathway is crucial for tilapia T cells to exert proper immune functions.

Next, we investigated the role of CTRP9 in mediating AdipoR1–CaMKKβ axis-dependent antibacterial T-cell responses. Upon *A. hydrophila* infection, *CTRP9* transcription was

significantly upregulated in spleen leukocytes (Appendix Fig. S8A), suggesting its potential involvement in host antibacterial immunity. To assess its functional relevance, *CTRP9* expression was silenced using siRNA during *A. hydrophila* infection (Appendix Fig. S8B). *CTRP9* knockdown markedly suppressed the infection-induced upregulation of *AdipoR1*, *CaM*, *CaMKKβ*, and *AMPK* (Appendix

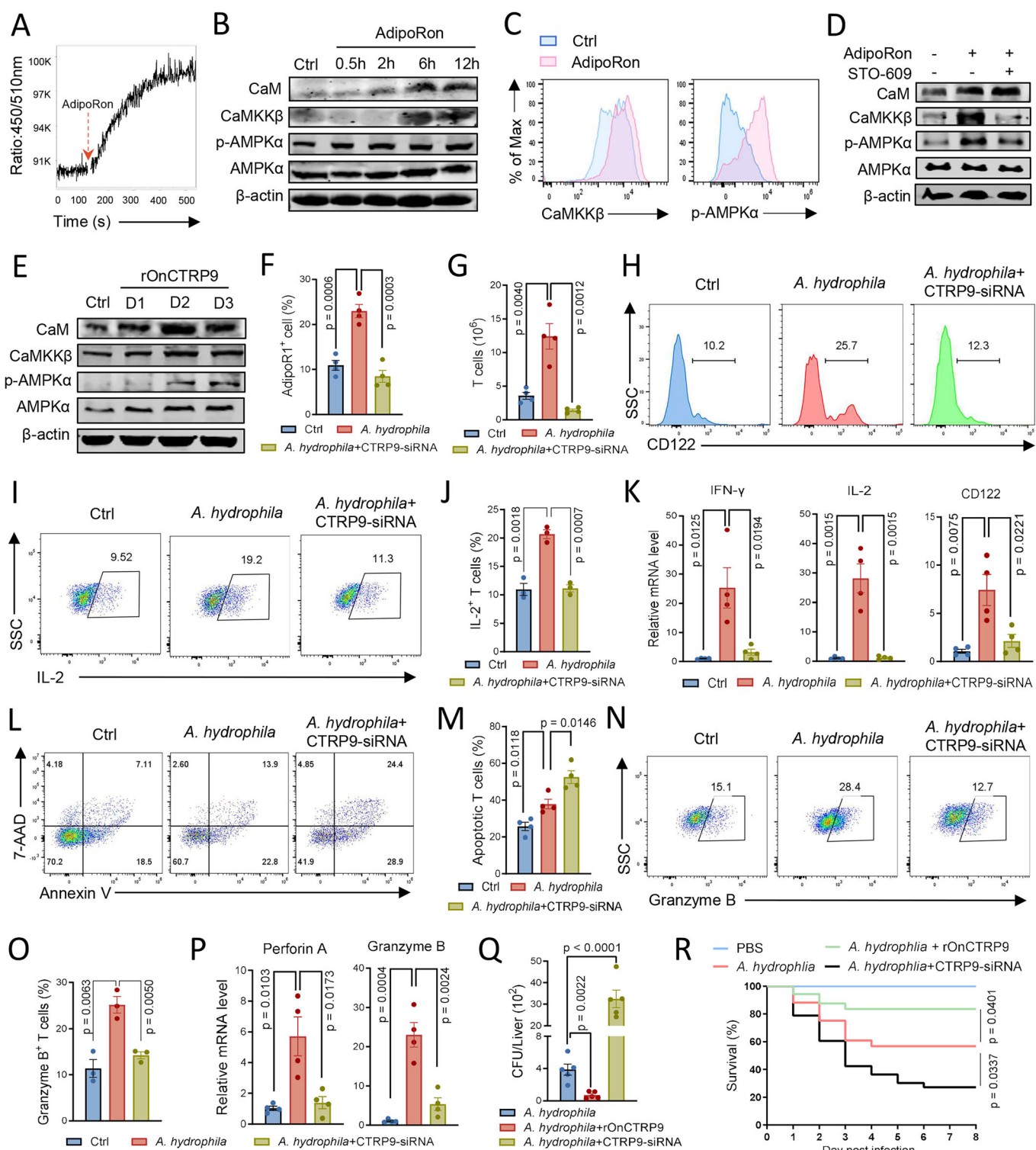

Fig. S8C), confirming the essential role of CTRP9 in initiating the AdipoR1–Ca²⁺–CaMKKβ signaling cascade in response to bacterial challenge. Functionally, *CTRP9* silencing impaired the expansion of AdipoR1⁺ lymphocytes (Fig. 4F; Appendix Fig. S8D) and CD3⁺ T cells (Fig. 4G; Appendix Fig. S8E,F), decreased the proportion of CD122⁺ T cells (Fig. 4H), and reduced IL-2-producing T cells

(Fig. 4I,J). In addition, the expression of *IFN-γ*, *CD122*, and *IL-2* was significantly diminished (Fig. 4K), while T-cell apoptosis was exacerbated (Fig. 4L,M). Moreover, *CTRP9* knockdown reduced the frequency of Granzyme B-producing T cells (Fig. 4N,O), impaired the transcriptional upregulation of *Perforin A*, *Granzyme B*, *TNF-α*, and *IL-6* (Fig. 4P; Appendix Fig. S8G), ultimately resulting in

**Figure 4.  CTRP9 activates a Ca²⁺–CaMKKβ–AMPK axis via AdipoR1 in tilapia T cells to resist bacterial infection.**

(A) Ca²⁺ influx of Indo-1–loaded spleen lymphocytes was determined by flow cytometry based on the change in the 450:510 nm ratio after AdipoRon stimulation. (B, C) Spleen leukocytes were stimulated with AdipoRon. Western blot assay showing the protein or phosphorylation levels of CaM, CaMKKβ, AMPKα at the indicated times (B). Overlaid histograms showing the expression levels of indicated molecules in gated CD3⁺ T cells at 6 h after stimulation (C). (D) Spleen leukocytes isolated from tilapia were stimulated with AdipoRon in the presence or absence of CaMKKβ inhibitor STO-609, western blot assay showing the protein or phosphorylation levels of indicated molecules at 6 h after stimulation. (E) Tilapia individuals were *i.p.* injected with or without rOnCTRP9 on days 0 and 1, and the spleen leukocytes were harvested on days 1, 2, and 3 for assay. Western blot showing the protein or phosphorylation levels of CaM, CaMKKβ, AMPKα at indicated times. (F–R) Tilapia *i.p.* injected with CTRP9-specific or control siRNA were infected with *A. hydrophila*, and spleen leukocytes were harvest on 5 dpi for assay. (F) Bar figure showing the percentage of AdipoR1⁺ T cells in gated lymphocytes, n = 4. (G) The absolute number of CD3⁺ T cells are shown, n = 4. (H) Histograms showing the levels of CD122 in gated CD3⁺ T cells. (I, J) Tilapia individuals were *i.p.* injected with BFA 6 h before sacrifice. Representative FACS plots (I) and bar figure (J) showing the percentage of IL-2⁺ T cells in gated CD3⁺ T cells, n = 3. (K) Relative mRNA levels of *CD122*, *IL-2*, and *IFN-γ* by qPCR, n = 4. (L, M) Representative FACS plots (L) and bar figure (M) showing the 7-AAD and Annexin V staining on gated CD3⁺ T-cell population, n = 4. (N, O) Tilapia individuals were *i.p.* injected with BFA 6 h before sacrifice, representative FACS plots (N) and bar figure (O) showing the percentage of Granzyme B⁺ T cells in gated CD3⁺ T cells, n = 3. (P) Relative mRNA levels of *Perforin A* and *Granzyme B* by qPCR, n = 4. (Q) *A. hydrophila* titers in the liver of infected tilapia on 5 dpi, n = 5. (R) Kaplan–Meier survival plot showing the survival percentage of tilapia, n = 25. Data information: n stands for biological replicates. Error bars indicate mean ± SEM. Significance between the groups was determined by a two-tailed Student's t test.

increased susceptibility to bacterial infection (Fig. 4Q) and elevated mortality (Fig. 4R). Collectively, these results demonstrate that CTRP9 is a critical initiator of the AdipoR1–Ca²⁺–CaMKKβ axis in tilapia T cells and plays an indispensable role in mounting effective antibacterial immune responses.

## CTRP9 enhances T-cell immunity in tilapia by promoting glycolysis

We sought to investigate how CTRP9 regulates the T-cell response in tilapia. During *A. hydrophila* infection, rOnCTRP9 administration significantly promoted the pathogen-induced T-cell activation, as evidenced by the expansion of IL-2-producing T cells (Fig. 5A,B) and increased CD122 expression in T cells (Appendix Fig. S9A). In addition, rOnCTRP9 enhanced the ability of T cells to produce Granzyme B during infection (Fig. 5C,D). This enhancement of T-cell activation and function by CTRP9 is a direct effect that does not rely on the involvement of other cell lineages, as in vitro administration of rOnCTRP9 was sufficient to further augment TCR-induced upregulation of activation markers *IL-2*, *CD122*, and *CD44* (Fig. 5E), as well as effector molecules *Perforin A*, *Granzyme B*, and *IFN-γ* (Fig. 5F) in sorted CD3⁺ T cells. Furthermore, in vitro treatment with rOnCTRP9 reduced T-cell apoptosis (Appendix Fig. S9B,C), and alleviated apoptosis induced by lower energy (Fig. 5G,H). This protective effect was associated with decreased cleavage of Caspase-8 and Caspase-3 (Fig. 5I,J; Appendix Fig. S9D,E), suggesting that CTRP9 promotes T-cell survival by inhibiting caspase-dependent apoptotic pathways. These immuno-protective properties enable the administration of rOnCTRP9 during bacterial infection to enhance pathogen clearance (Fig. 4Q) and improve the survival of tilapia (Fig. 4R). Moreover, treatment of rOnCTRP9 increased the expression of glycolysis-related molecules Glut1, HK2, and PKM (Fig. 5K,L), enhanced glucose uptake (Fig. 5M), and accelerated extracellular acidification (ECAR) (Fig. 5N), indicating that CTRP9 promotes the T-cell glycolysis in tilapia. To confirm whether CTRP9 regulates T-cell glycolysis through AdipoR1 signaling, the AdipoR1 antibody was administered to tilapia during *A. hydrophila* infection. Treatment of rOnCTRP9 improved glucose uptake in CD3⁺ T cells, while this enhancement was significantly inhibited when AdipoR1 was blocked (Fig. 5O,P). Furthermore, blocking AdipoR1 using an antibody or inhibiting AMPK using an inhibitor reduced the expression of glycolysis-related molecules at both mRNA and

protein levels (Fig. 5Q,R; Appendix Fig. S9F), suggesting that CTRP9 promotes T-cell glycolysis via AdipoR1–AMPK signaling. Overall, our findings support the notion that the CTRP9–AdipoR1 axis promotes T-cell glycolysis and immunity in tilapia.

## CTRP9 promotes T-cell response in mouse

Although CTRP9 is known as an AdipoR ligand in mouse (Kambara et al, 2015), whether it regulates mammalian T-cell immunity remains unknown. We prepared a recombinant mouse CTRP9 protein (mCTRP9; Appendix Fig. S10A) and investigated its regulatory effects on mouse T-cell response. Similar to the observations in tilapia, mCTRP9 treatment induced the expression of CaM and CaMKKβ and enhanced AMPKα phosphorylation in mouse splenocytes (Fig. 6A). Blocking CaMKKβ significantly impaired the CTRP9-induced AMPK phosphorylation (Fig. 6A), suggesting that mouse CTRP9 also triggers the Ca²⁺–CaMKKβ–AMPK pathway. In addition, the mCTRP9 treatment of mouse splenocytes markedly elevated the phosphorylation of JNK, AKT, ERK1/2, NF-κB p65, and S6 (Fig. 6B), indicating enhanced T-cell activation. This was further confirmed by a T-cell activation assay, where mCTRP9 administration increased CD3/CD28 mAb-induced CD25 and CD69 expression in T cells, especially in CD4⁺ T cells (Fig. 6C,D). Consistent with this finding, mCTRP9 conferred both CD4⁺ and CD8⁺ T cells with a stronger proliferation ability upon CD3/CD28 mAb stimulation (Fig. 6E). Regarding cytokine production, mCTRP9 significantly enhanced IL-2 production in CD4⁺ T cells upon P + I or CD3/CD28 mAb stimulation (Fig. 6F–H; Appendix Fig. S10B, top panels), but had no effect on or inhibited IL-2 production in CD8⁺ T cells (Fig. 6F–H; Appendix Fig. S10B, bottom panels). The same trend was observed for IFN-γ production (Fig. 6I–K; Appendix Fig. S10C). Compared with P + I group, P + I stimulation with additional mCTRP9 induced higher expression of T-bet and NF-κB—the key transcription factors for IFN-γ and IL-2 respectively—in CD4⁺ T cells (Fig. 6L–O, left panels), In contrast, CTRP9-induced upregulation of transcription factors cannot be observed in CD8⁺ T cells (Fig. 6L–O, right panels), suggesting CTRP9 differently regulates the cytokine production of CD4⁺ and CD8⁺ T cells via inducing different expression patterns of transcription factor. Overall, we demonstrated, for the first time to our knowledge, that CTRP9 enhances the activation and proliferation of both CD4⁺ and CD8⁺ T cells and improves the cytokine production ability of CD4⁺ T cells in mice.

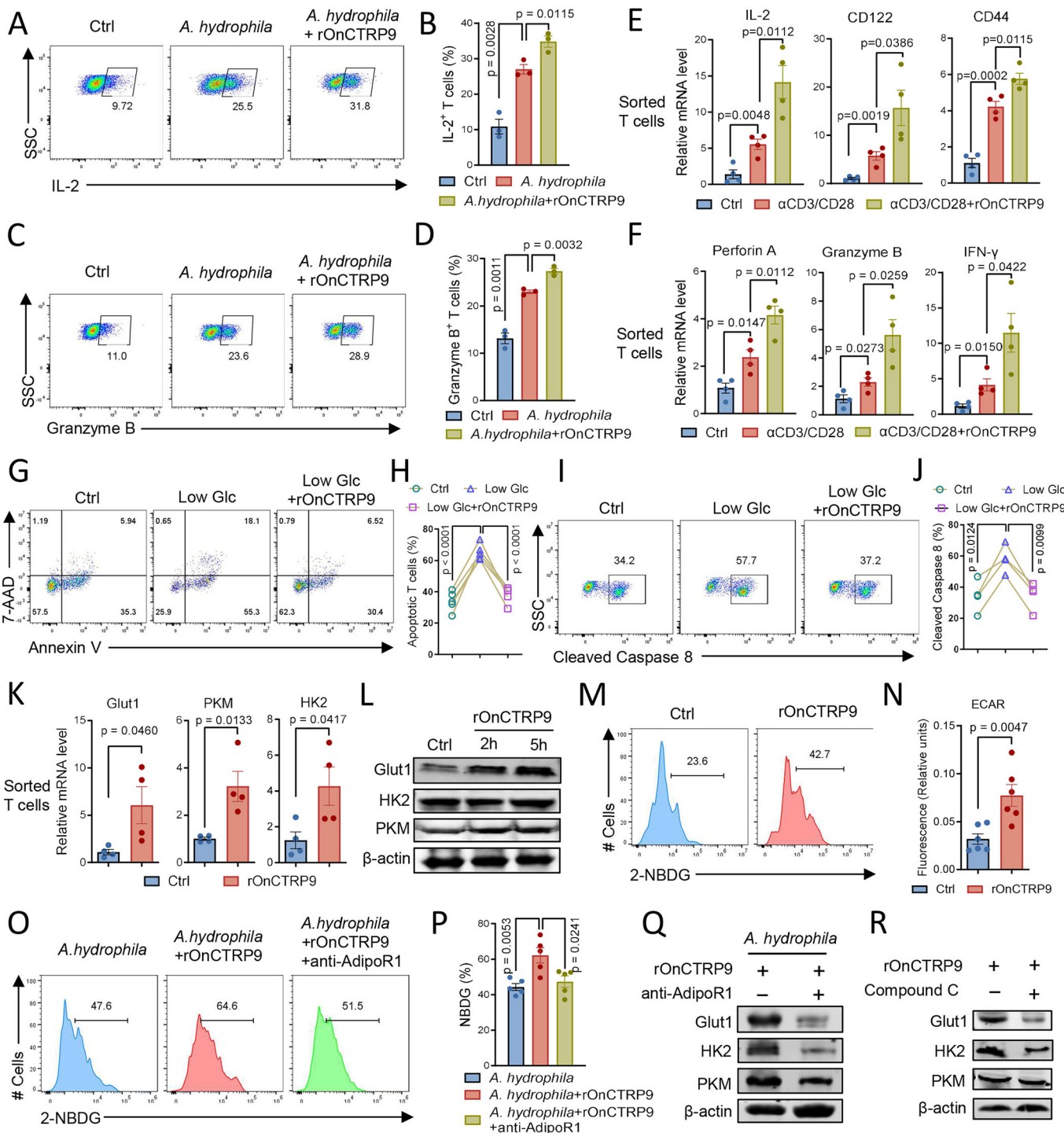

## CTRP9 enhances the activity of CD19-targeting CAR-T cells against B-cell lymphoma

Given that CTRP9 enhances T-cell responses in mice, we sought to determine whether this adipokine could optimize T-cell-mediated cancer immunotherapy. A retroviral plasmid encoding mouse CD19 CAR was constructed (Fig. 7A), and used to infect mouse T cells, thereby generating CAR-T cells targeting CD19+ cells.

These anti-CD19 CAR-T cells were incubated with CD19+ A20 B-cell lymphoma in vitro, either in the presence or absence of mCTRP9; then we evaluated target cell apoptosis, cytokine production in CAR-T cells, and cytotoxicity (Fig. 7B; Appendix Fig. S11). T cells infected with the control retrovirus had minimal effects on inducing apoptosis in A20 B-cell lymphoma (Fig. 7C). In contrast, anti-CD19 CAR-T cells effectively killed the target cells, while mCTRP9 further enhanced CAR-T cell-mediated target cell

**Figure 5. CTRP9 enhances T-cell immunity of tilapia by promoting glycolysis.**

(A–D) Tilapia individuals that infected with *A. hydrophila* were *i.p.* injected with or without rOnCTRP9 on days 1, 2, and 4, and were *i.p.* injected with BFA 6 h before sacrifice. Spleen leukocytes were isolated on 5 dpi. Representative FACS plots (A, C) and bar figure (B, D) showing the percentage of IL-2$^+$ T cells or Granzyme B$^+$ T cells in gated CD3$^+$ T cells, $n = 3$. (E, F) Spleen T cells sorted from healthy tilapia were stimulated with mouse anti-tilapia CD3ε and CD28 mAb in the presence or absence of rOnCTRP9 for 6 h, and relative mRNA levels of the indicated molecules were examined by qPCR, $n = 4$. (G–J) Spleen leukocytes that cultured in the medium containing normal glucose (4.5 g/L) and low glucose (1.125 g/L) were treated with rOnCTRP9 or not for 12 h. (G, H) Representative FACS plots (G) and statistical figure (H) showing the 7-AAD and Annexin V staining on gated CD3$^+$ T-cell population, $n = 5$. (I, J) Representative FACS plots (I) and bar figure (J) showing the percentage of cleaved caspase 8 in gated CD3$^+$ T cells, $n = 4$. (K–N) Spleen leukocytes or sorted T cells were treated with rOnCTRP9 or not. (K) Relative mRNA levels of the indicated molecules were examined by qPCR in sorted T cells at 6 h, $n = 4$. (L) Western blot showing the protein levels of the indicated molecules at the indicated time points in leukocytes. (M) Glucose uptake in gated CD3$^+$ T cells was measured by 2-NBDG$^+$ cells at 5 h after treatment. (N) Extracellular acidification rate (ECAR) after 3 h of CTRP9 treatment, $n = 6$. (O–Q) Tilapia individuals infected with *A. hydrophila* were *i.p.* injected with or without rat anti-tilapia AdipoR1 antibody on days 1 and 3, or rOnCTRP9 on days 1, 2, and 4. Spleen leukocytes were isolated on 5 dpi. (O, P) Histograms (O) and bar figure (P) showing glucose uptake in gated CD3$^+$ T cells, $n = 5$. (Q) Western blot showing protein levels of the indicated molecules. (R) Spleen leukocytes stimulated with rOnCTRP9 were treated with 10 μM Compound C or not for 6 h, and western blot showing the protein levels of Glut1, HK2, and PKM. Data information: $n$ stands for biological replicates. Error bars indicate mean ± SEM. Significance between the groups was determined by a two-tailed Student's *t* test.

death (Fig. 7D,E). This was corroborated by the increased cleavage of caspase-3 in A20 lymphoma (Fig. 7F,G). Moreover, mCTRP9 treatment upregulated the production of IL-2 (Fig. 7H,I), IFN-γ (Fig. 7J,K), TNF-α (Fig. 7L,M), and Granzyme B (Fig. 7N,O) in both CD4$^+$ and CD8$^+$ subpopulations of anti-CD19 CAR-T cells, indicating that CTRP9 may promote CAR-T-cell functionality. This was further validated by cytotoxicity assays. In the presence of mCTRP9, anti-CD19 CAR-T cells exhibited significantly CD107a degranulation (Fig. 7P,Q) and stronger cytolytic activity (Fig. 7R) against B-cell lymphoma, suggesting an enhanced cytotoxicity. Overall, our findings demonstrate that the adipokine CTRP9 has the potential to serve as a promising adjuvant to enhance the efficacy of CAR-T-cell-mediated cancer immunotherapy.

## CTRP9 and ADPN differentially regulate T-cell glycolysis in mouse

As the canonical AdipoR ligand, ADPN, suppresses T-cell immunity (Li et al, 2019; Wilk et al, 2011; Xiao et al, 2017), whereas another ligand, CTRP9, demonstrates the opposite effect in this study, we sought to further elucidate the mechanism underpinning this differential modulation. Considering that tilapia CTRP9 enhances glycolysis—a metabolic pathway essential for T-cell function—we conducted a comparative analysis of how these adipokines regulate T-cell glycolysis in mouse. In resting splenocytes, neither CTRP9 nor ADPN treatment caused significant changes in AdipoR1 at mRNA and protein levels (Appendix Fig. S12A,B). However, mCTRP9 treatment significantly upregulated the expression of Glut1 and key glycolytic enzymes HK2, PKM and PFKP, whereas ADPN elicited no effect or slight inhibitory effect (Fig. 8A). At the protein level, CTRP9's stimulatory effects and ADPN's inhibitory effects were more pronounced (Fig. 8B). Consistently, CTRP9 enhanced glucose uptake in CD3$^+$ T cells, while ADPN treatment impaired this capacity (Fig. 8C). These results indicate that CTRP9 and ADPN differentially modulate glycolysis in resting splenocytes, which was further corroborated by the activity of HK2 and PFK, although ADPN only mildly suppressed these enzyme activities (Fig. 8D). We suspected that the partial inhibition of glycolysis by ADPN was due to a relative lower level of glycolysis in naive T cells, therefore we assessed the regulatory effects of ADPN on glycolysis in the activated T cells, since T cells substantially upregulate glycolysis upon activation to fulfill the demand of proliferation and effector

function (Chang et al, 2013; Levine et al, 2021). Splenocytes of the mouse were stimulated with CD3 and CD28 mAb. T-cell activation resulted in a consistent upregulation of glycolysis-related genes at both the mRNA and protein levels; however, ADPN impaired this upregulation (Fig. 8E,F). Activated CD4$^+$ and CD8$^+$ T cells exhibited enhanced glucose uptake; in contrast, ADPN exerted an inhibitory effect on T-cell activation-induced glucose uptake (Fig. 8G,H). In line with this, the glycolysis-related enzyme activity induced by T-cell activation was also suppressed by additional ADPN (Fig. 8I). Therefore, we confirmed that ADPN could inhibit T-cell glycolysis. Considering the pivotal role of glycolysis in T-cell function, we propose that the differential regulation of glycolysis by CTRP9 and ADPN contributes to their distinct effects on T-cell immune responses.

## Discussion

To date, the understanding of AdipoR1 has been largely restricted to its role in regulating energy and metabolism. Recent studies in mammals suggest that AdipoR1 is involved in T-cell immunity (Shibata et al, 2015; Zhang et al, 2020), indicating that crosstalk occurs between AdipoR1 signaling and T-cell responses. Nevertheless, the relationship between AdipoR and T-cell lineages remains unclear. Notably, AdipoR1 signaling appears to differentially regulate the same T-cell-related immunological events such as Th17 differentiation (Li et al, 2019; Xiao et al, 2017; Zhang et al, 2020), but the nature of this diversity is not yet understood. In addition, it remains to be determined whether the coordination between AdipoR and T-cell immunity was independently acquired in mammals or gradually evolved in vertebrates. In the present study, we used tilapia and mouse models to address these questions, providing novel insights into the immunological functions and evolutionary significance of AdipoR signaling.

AdipoR1 is an evolutionarily ancient receptor. The AdipoRon, a commercial agonist developed for mammalian AdipoR, effectively activated AdipoR1 in tilapia, which provided additional evidence for its conservation. Although recent studies have indicated the involvement of AdipoR in T-cell response, its relationship with T-cell lineages has remained largely unexplored. We found that the majority of spleen T cells in tilapia expressed AdipoR1, and most spleen AdipoR1$^+$ cells were indeed T cells, which revealed—for the first time—a strong association between AdipoR and T cells.

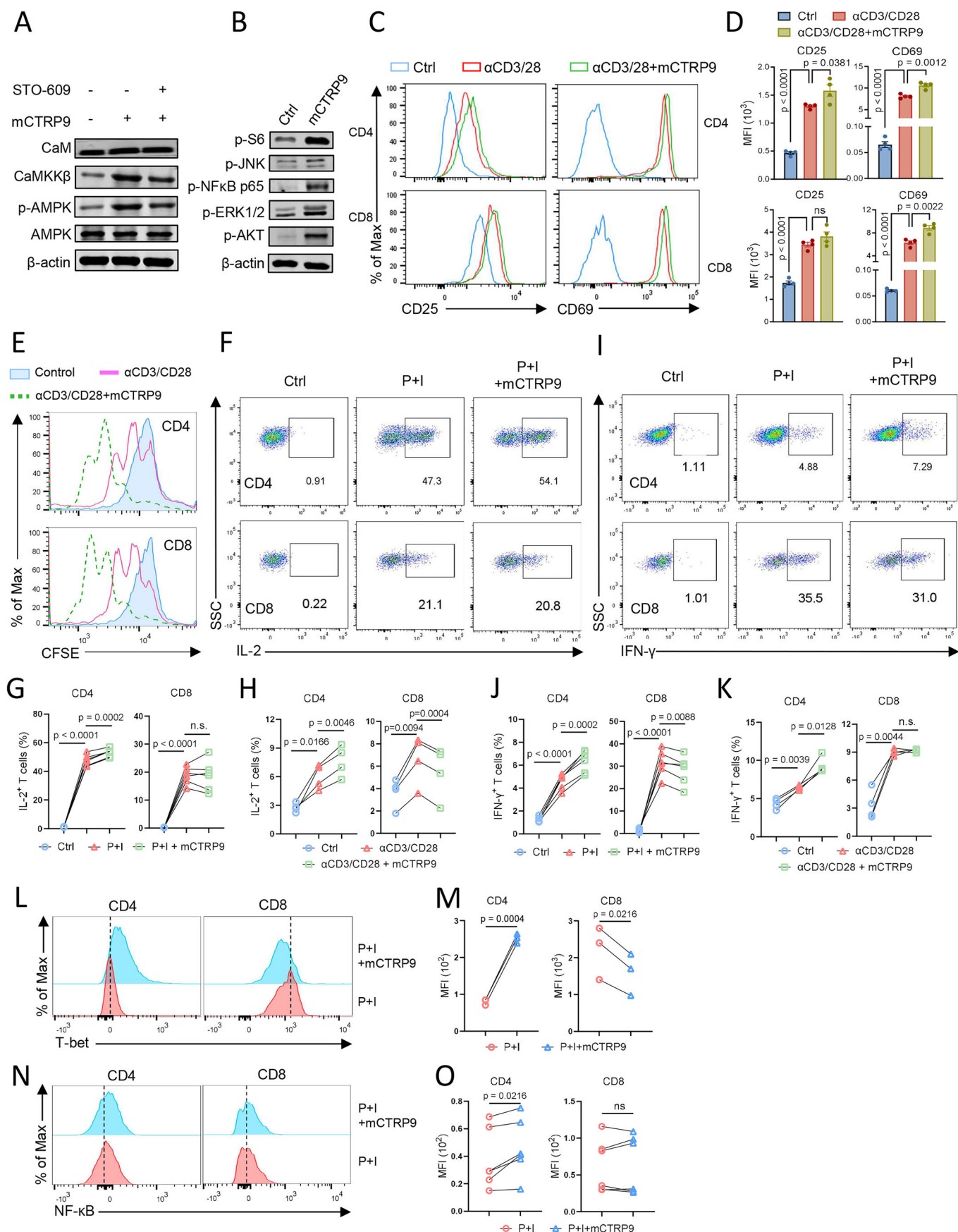

**Figure 6. CTRP9 triggers AdipoR signal and promotes T-cell immunity in mouse.**

(A, B) Mouse splenocytes were stimulated with mCTRP9 in the presence or absence of STO-609 for 6 h, and western blot showing protein or phosphorylation levels of the indicated molecules. (C–O) Mouse splenocytes were stimulated with anti-mouse CD3 plus anti-mouse CD28 (C–E, H, K) or P + I (F, G, I, J, L–O) in the presence or absence of mCTRP9. Overlaid histograms (C, L, N) and bar figures (D, M, O) showing the expression levels of indicated molecules in gated CD4$^+$ T cells or CD8$^+$ T cells at 12 h (C, D) or 4 h (L–O) post stimulation. Proliferation of CFSE-labeled CD4$^+$ and CD8$^+$ T cells was examined by flow cytometry at 72 h post stimulation (E). Representative FACS plots (F, I) and statistical figures (G, H, J, K) showing the percentage of IL-2$^+$ cells and IFN-γ$^+$ cells in gated CD4$^+$ or CD8$^+$ T-cell population at 5 h (F, G, I, J) or 24 h (H, K) post stimulation. $n = 6$ (G, J, O), $n = 4$ (H, K), or $n = 3$ (M). Data information: $n$ stands for biological replicates. Error bars indicate mean ± SEM. Significance between the groups was determined by a two-tailed Student's $t$ test.

Recently, metabolic programs have been recognized as key determinants of T-cell immune function and fate, with several transcription factors, signaling pathways, and nutrient transporters identified to be associated with T-cell immune regulation (Chapman et al, 2020; Huang et al, 2021; Klein Geltink et al, 2020; Lim et al, 2022). Considering the high expression level of the metabolism-related receptor AdipoR1 in T cells, we speculate that AdipoR1 is a previously unknown hub coupling metabolism and T-cell immunity.

Ca$^{2+}$ extensively regulates various cellular processes. In adipocytes, downstream of AdipoR1, Ca$^{2+}$ signaling is integrated with AMPK through the CaM–CaMKKβ axis (Lin et al, 2011; Iwabu et al, 2010), which mediates glucose and lipid metabolism and maintains energy balance (Meier and Gressner, 2004). In contrast, in T cells, TCR signaling induces Ca$^{2+}$ influx, activating CaN or CaMKKβ, which in turn regulates the nuclear translocation of NFAT or CREB for T-cell proliferation (Oh-hora, 2009; Wei et al, 2020a). Thus, adipocytes and T cells employ AdipoR1 and TCR, respectively, to trigger the Ca$^{2+}$–CaM–CaMKKβ axis. To our knowledge, this is the first study to demonstrate that AdipoR1 is also crucial for T cells to initiate Ca$^{2+}$ influx and CaM–CaMKKβ signaling, indicating a potential crosstalk between AdipoR1 signaling and TCR signaling. Considering AMPK inhibition hinders T-cell responses in tilapia (Li et al, 2023c), we demonstrate that AdipoR1 activates the CaM–CaMKKβ–AMPK pathway through Ca$^{2+}$ influx, thus ensuring proper immune function in fish T cells. These results provide a reasonable explanation for the effect of AdipoR1 on T-cell responses, indicating that AdipoR1 and TCR share downstream Ca$^{2+}$ signaling, enhancing the diversity and plasticity of T-cell regulation. Moreover, the involvement of AMPK, a central player in energy and metabolism, reinforces the notion that AdipoR1 is a novel checkpoint coupling metabolism and immune response.

AdipoR has been reported to extensively regulate the innate immunity. AdipoR1 modulates the production of IL-6 via downstream AMPK signaling (Tang et al, 2007) and inhibits cyclooxygenase COX2, a target gene of NF-κB, to promote inflammation (Chandrasekar et al, 2008; Prasatthong et al, 2021). AdipoR also regulates the expression of inflammatory cytokines and hemocyte apoptosis in invertebrates such as oysters (Ge et al, 2020). In contrast, fewer studies investigate its role in adaptive immune responses, and the existing findings are controversial and contradictory. Some studies support the notion that AdipoR1 signaling negatively regulates T-cell immunity by promoting the development of Treg cells and IL-10 secretion (Ramos-Ramírez et al, 2021) while inhibiting Th17 differentiation by suppressing glycolysis (Surendar et al, 2019). However, other studies suggested that deficiency of AdipoR1 also impairs glycolysis-driven Th17

differentiation (Zhang et al, 2020). In the early vertebrate Nile tilapia, AdipoR1 contributes to T-cell proliferation, survival, and effector functions, thus playing a crucial role in the antibacterial immune response. This finding supports the positive modulation of AdipoR1 on T-cell response in early vertebrate, and proposes that AdipoR-mediated regulation of T-cell immunity is an ancient survival strategy that was programmed before the evolutionary emergence of tetrapods.

We attempted to explain the differential regulation of T-cell immunity by AdipoR from the ligand perspective. The adipokine ADPN—a well-known ligand for AdipoR (Fang and Judd, 2018; Straub and Scherer, 2019)—suppresses T-cell immunity through AdipoR signaling (Shibata et al, 2015; Wilk et al, 2011). However, ADPN is not the only ligand for AdipoR; its homolog CTRP9 has also been identified as another ligand (Guan et al, 2022; Guo et al, 2022), although its role on T-cell immunity is unclear. In the present study, we found that all species within the genus Tilapia lack ADPN, while tilapia alternatively utilizes CTRP9 to bind AdipoR1, activate the CaM–CaMKKβ–AMPK signaling pathway, and promote glucose uptake and glycolysis in T cells, thereby enhancing T-cell survival. Indeed, this enhancement of T-cell immunity by CTRP9 is not restricted to early vertebrates. Our study in mice demonstrated that CTRP9 promotes the activation and proliferation of CD4$^+$ and CD8$^+$ T cells, while also increasing IL-2 and IFN-γ production in CD4$^+$ T cells. Given the positive effects of CTRP9 on T-cell immunity, it is unsurprising that we observed CTRP9 enhancing the cytokine production and cytotoxicity of anti-CD19 CAR-T cells in mice, thereby facilitating a more efficient eradication of CD19$^+$ B-cell lymphoma. Notably, in contrast to in vitro-activated primary T cells, CTRP9 enhanced cytokine production in both CD4$^+$ and CD8$^+$ CAR-T cells. We speculate that CAR-derived potent antigen-specific signals, combined with the complex immune synapse formed during CAR-T/ tumor cell interactions, collectively reprogram CD8$^+$ T-cell responsiveness and provide essential synergistic signaling. These differential effects establish that CTRP9-mediated immunomodulation of T-cell subsets is not static but exhibits microenvironment-dependent plasticity. To the best of our knowledge, this represents the first description that CTRP9 enhances the function of T cells and CAR-T cells in mammals, indicating CTRP9 may serve as a promising adjuvant to optimize the CAR-T cell-mediated cancer therapy. This finding stands in stark contrast to the effect of ADPN on T-cell immunity, as it inhibits T-cell activation, proliferation, cytokine production, and antitumor responses while enhancing the immunosuppressive function of Treg cells (Chikaishi et al, 2023; Li et al, 2019; Ramos-Ramírez et al, 2021). We therefore propose a novel perspective: AdipoR1 differentially regulates T-cell immune outcomes by binding to distinct ligands, ADPN and CTRP9.

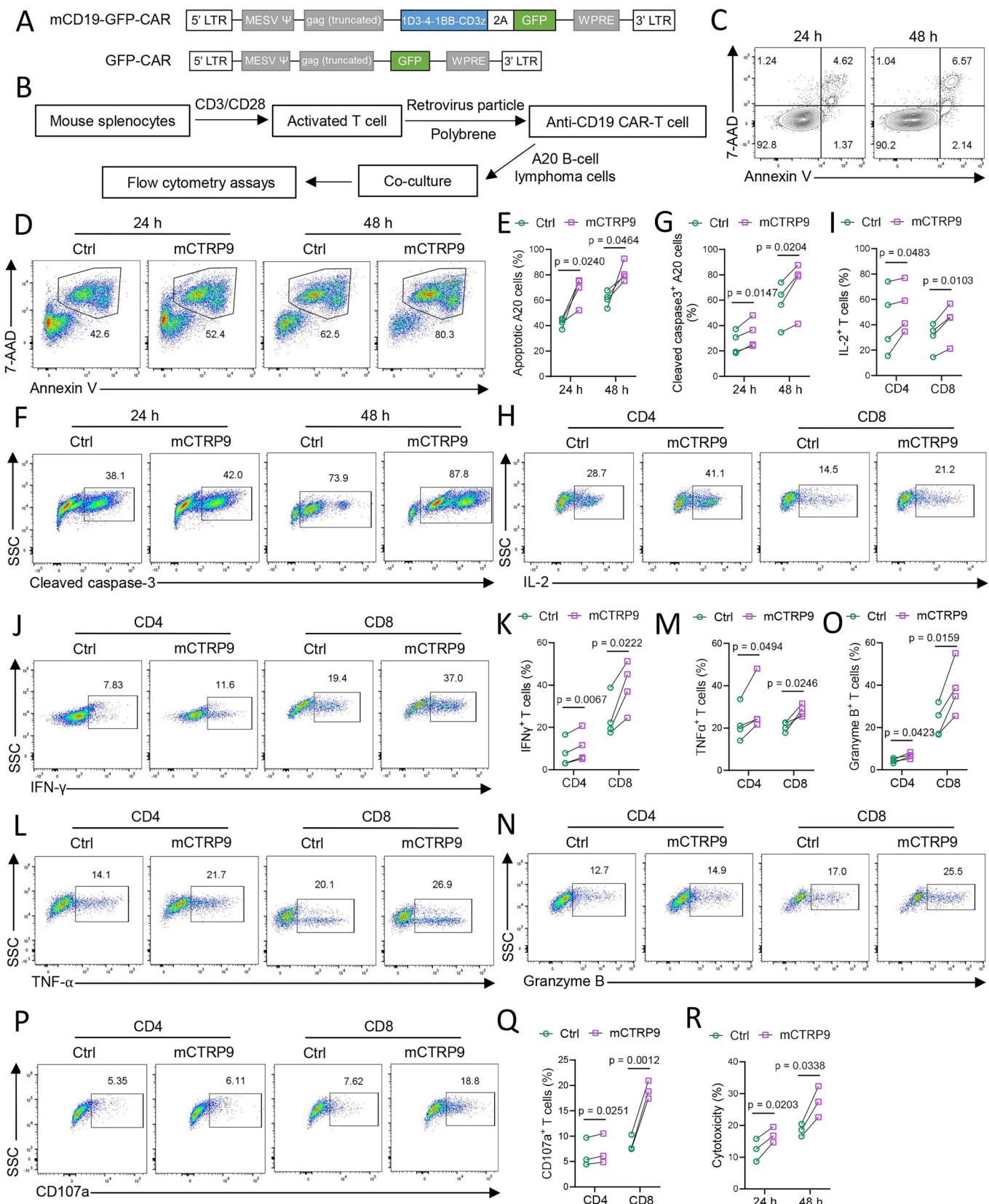

**Figure 7. CTRP9 enhances the activity of anti-CD19 CAR-T cells against B-cell lymphoma.**

(A) Construction of mouse CD19 CAR. (B) The anti-CD19 CAR-T cells were incubated with CD19$^+$ A20 B-cell lymphoma cells in the presence or absence of mouse CTRP9 in vitro for assay. (C–E) Representative FACS plots (C, D) and statistical figure (D) showing the 7-AAD and Annexin V staining in gated CD19$^+$ A20 B-cell lymphoma cells incubating with the control CAR-T cells (C) or anti-CD19 CAR-T cells (D, E), n = 4. (F, G) Representative FACS plots (F) and statistical figure (G) showing the cleaved caspase-3 staining in gated CD19$^+$ A20 B-cell lymphoma cells, n = 4. (H–P) Representative FACS plots (H, J, L, N, P) and statistical figures (I, K, M, O, Q) showing the percentage of IL-2$^+$, IFN-γ$^+$, TNF-α$^+$, Granzyme B$^+$ cells and CD107a$^+$ cells in gated CD4$^+$ or CD8$^+$ CAR-T cells at 24 h in the presence or absence of mouse CTRP9, n = 4. (R) Cytotoxicity of CAR-T cells was determined by LDH release at 24 h, n = 3. The paired values of Ctrl and mCTRP9 treatments refer to cells from the same donor mouse subjected to different treatments. Data information: n stands for biological replicates. Error bars indicate mean ± SEM. Significance between the groups was determined by a two-tailed Student's t test.

Upon activation, T cells undergo profound metabolic reprogramming. Naive T cells primarily rely on fatty acid oxidation and oxidative phosphorylation for energy production, whereas effector T cells rapidly upregulate glycolysis, glutamine metabolism, and fatty acid synthesis (Levine et al, 2021; Wang et al, 2011). Glycolysis not only fulfills the energy and biosynthetic demands of effector T cells but also contributes to the regulation of T-cell function and fate (Cao et al, 2023; Wenes et al, 2022); and deficiency or alteration in glycolysis is tightly associated with the T-cell dysfunction (Cao et al, 2023; Zhao et al, 2016). In the mouse model, we demonstrated that CTRP9 significantly enhances glycolytic activity in T cells, aligning with its role in augmenting T-cell functionality. Conversely, ADPN was found to suppress glucose uptake and glycolysis in mouse T cells. We speculated that the opposing effects of these two ligands might result from distinct conformational changes in AdipoR1 upon ligand engagement, as similar mechanisms have been observed in TNF receptors (TNFRs) and G protein-coupled receptors (GPCR) (Heydenreich et al, 2023; Murali et al, 2005). For example, TNFR1 forms a trimeric complex upon binding to its natural ligand TNF-α, which attenuates TNF-α-induced NF-κB and p38 MAPK signaling, thereby reducing arthritis severity in mice. Notably, this interaction does not involve significant conformational changes in TNFR1. In contrast, binding of a pseudoallosteric ligand to TNFR1 induces a shift in the spatial orientation of tryptophan-107, which disrupts receptor signaling and renders TNFR1 functionally inactive (Murali et al, 2005). Although the precise mechanisms underlying the opposing effects mediated by AdipoR1 remain to be fully elucidated, current evidence highlights the differential regulation of T-cell glycolysis by CTRP9 and ADPN as a key determinant of their distinct impacts on T-cell immune responses. In addition, we speculated that this discrepancy in T-cell immunity may also be attributed to the involvement of the downstream AMPK pathway of AdipoR, considering its complexity and diversity in determining T-cell metabolism, function, and fate (He et al, 2021; Ma et al, 2017; Mamedov et al, 2023). It would be interesting to examine how these two adipokines coordinate or cooperate to regulate T-cell immunity via AdipoR in animals that possess both ADPN and CTRP9, such as non-cichlid fish and mouse.

In summary, we demonstrated that Nile tilapia AdipoR1 is closely associated with the T-cell lineage. Upon bacterial infection, AdipoR1 mediates Ca$^{2+}$ influx and activates the CaM–CaMKKβ–AMPK pathway, facilitating crosstalk with TCR signaling. This cascade ultimately promotes T-cell activation, proliferation, and antimicrobial immunity through enhanced glycolysis (Fig. 8J). Although tilapia lacks ADPN, it relies on CTRP9 to activate AdipoR1 signaling and bolster T-cell immunity. Notably, CTRP9 also enhances T-cell activation, proliferation, and cytokine production in mice (Fig. 8J), and boosts the activity of anti-CD19 CAR-T cells against B-cell lymphoma. These findings uncover a previously unknown, yet evolutionarily conserved mechanism underlying T-cell immunity. Importantly, the two adipokine ligands of AdipoR—CTRP9 and ADPN—exert opposing effects on T-cell glycolysis in mice, with CTRP9 promoting and ADPN inhibiting glycolysis (Fig. 8J). This suggests that AdipoR1 differentially regulates T-cell immunological outcomes through binding to distinct adipokines, potentially resolving the seemingly contradictory observations regarding AdipoR's regulation of T cells. Our findings provide a novel perspective on how metabolic and adipocyte signals modulate T-cell immunity.

## Methods

**Reagents and tools table**

| Reagent/resource | Reference or source | Identifier or catalog number |
|---|---|---|
| **Experimental models** | | |
| Nile tilapia | Guangzhou, Guangdong Province, China | N/A |
| BALB/c mice | East China Normal University | N/A |
| *Aeromonas hydrophila* | Li et al, 2023a | N/A |
| A20 B-cell lymphoma cells | ATCC | TIB-208 |
| Anti-CD19 CAR-T cells | Wenzhou Medical University | N/A |
| BOSC23 cells | Wenzhou Medical University | N/A |
| **Recombinant DNA** | | |
| pET-28a | Ai et al, 2022 | N/A |
| MSCV plasmid | Wenzhou Medical University | N/A |
| mCD19-GFP-CAR plasmid | Wenzhou Medical University | N/A |
| GFP-CAR plasmid | Wenzhou Medical University | N/A |
| PCL-ECO helper plasmid | Wenzhou Medical University | N/A |

| Reagent/resource | Reference or source | Identifier or catalog number |
|---|---|---|
| **Antibodies** | | |
| AP-conjugated goat anti-rat IgG | Solarbio | Cat #K1032G-AP |
| Anti-AdipoR1 polyclonal antibody | This study | N/A |
| Anti-mouse CD3 | BioLegend | Cat #100301 |
| Anti-mouse CD28 | BioLegend | Cat #102101 |
| Anti-tilapia CD3 | Li et al, 2023b | N/A |
| Anti-tilapia CD28 | Li et al, 2023b | N/A |
| Anti-tilapia CD122 | Geng et al, 2024 | N/A |
| Alexa Fluor 647-conjugated goat anti-mouse IgG H&L | Abcam | Cat #ab150115 |
| FITC-conjugated mouse anti-tilapia CD3ε | Li et al, 2023b | N/A |
| FITC-conjugated anti-mouse CD4 | BioLegend | Cat #100509 |
| PE-conjugated anti-mouse CD8α | BioLegend | Cat #162303 |
| APC-conjugated anti-mouse CD69 | BioLegend | Cat #104513 |
| PerCP/Cyanine 5.5-conjugated anti-mouse CD25 | BioLegend | Cat #101911 |
| Alexa Fluor 647-conjugated goat anti-rat | Abcam | Cat #ab150159 |
| PE-Cy7-conjugated anti-mouse CD19 | BioLegend | Cat #115520 |
| BV421-conjugated anti-mouse CD107a | BioLegend | Cat #121617 |
| Anti-tilapia IL-2 | Zhang et al, 2024a | N/A |
| Anti-tilapia granzyme B | Cao et al, 2024 | N/A |
| Anti-cleaved caspase 3 | Cell Signaling Technology | Cat #9664 |
| Anti-cleaved caspase 8 | Cell Signaling Technology | Cat #9496 |
| Anti-p-AMPKα | Cell Signaling Technology | Cat #2535 |
| Anti-CaM | Cell Signaling Technology | Cat #4830 |
| Anti-CaMKKβ | Cell Signaling Technology | Cat #16810 |
| APC-conjugated anti-mouse IL-2 | BioLegend | Cat #503809 |
| PE-conjugated anti-mouse TNF-α | BioLegend | Cat #506305 |
| Brilliant Violet 421-conjugated anti-mouse IFN-γ | BioLegend | Cat #505829 |
| PE anti-human/mouse Granzyme B Recombinant | BioLegend | Cat #372207 |
| Alexa Fluor 647-conjugated Annexin V | BioLegend | Cat #640920 |
| 7-AAD | Invitrogen | Cat #A1310 |
| APC-conjugated anti-BrdU | Invitrogen | Cat #17-5071-42 |

| Reagent/resource | Reference or source | Identifier or catalog number |
|---|---|---|
| Anti-p-S6 | Cell Signaling Technology | Cat #5364 |
| Anti-p-ERK1/2 | Cell Signaling Technology | Cat #4370 |
| Anti-p-AKT | Cell Signaling Technology | Cat #13038 |
| Anti-AMPKα | Cell Signaling Technology | Cat #5831 |
| Anti-β-actin | Cell Signaling Technology | Cat #4970 |
| Anti-p-JNK | Beyotime | Cat #AF1762 |
| Anti-p-p65 | Beyotime | Cat #AF5881 |
| Anti-Glut1 | Beyotime | Cat #AF1015 |
| Anti-HK2 | Beyotime | Cat #AG2142 |
| Anti-PKM | Beyotime | Cat #AF7764 |
| Anti-AdipoR1 | Beyotime | Cat #AF2131 |
| Alexa Fluor 800-conjugated goat-anti-rabbit IgG H&L | Cell Signaling Technology | Cat #5151 |
| Alexa Fluor 680-conjugated goat-anti-rat or mouse IgG H&L | Abcam | Cat #ab175775 |
| Goat Anti-Rat IgG H&L (Alexa Fluor 488) | Abcam | Cat #ab150157 |
| **Oligonucleotides and other sequence-based reagents** | | |
| PCR primers | This study | Appendix Table S2 |
| CTRP9 siRNA | This study | Appendix Table S2 |
| **Chemicals, enzymes, and other reagents** | | |
| BCA Protein Assay Kit | Sangon Biotech | Cat # C503021 |
| Compound C | MedChemExpress | Cat #BML-275 |
| Phorbol 12-myrustate 13-acetatae(PMA) | MedChemExpress | Cat #HY-18739 |
| Ionomycin | MedChemExpress | Cat #HY-13434 |
| BD cytofix/cytoperm buffer | BD Biosciences | Cat #554655 |
| BD perm/wash buffer | BD Biosciences | Cat #554723 |
| Indo-1 | Life Technologies | Cat #I1203 |
| 2-NBDG | Life Technologies | Cat #N13195 |
| Extracellular Acidification Rate (ECAR) Fluorometric Assay Kit | Elabscience | Cat #E-BC-F069 |
| BrdU | Sigma | Cat #19-160 |
| CFSE | Invitrogen | Cat #C34554 |
| Hoechst 33342 | Beyotime | Cat #C1022 |
| TRIzol | Invitrogen | Cat #A33251 |
| All-in-one 1st Strand cDNA Synthesis SuperMix | Novoprotein | Cat #E047 |
| NovoStart SYBR qPCR SuperMix Plus | Novoprotein | Cat #E096 |
| Phytohemagglutinin (PHA) | Sigma | Cat #11249738001 |
| AdipoRon | MedChemExpress | Cat #HY-15848 |
| STO-609 | MedChemExpress | Cat #HY-19805 |

| Reagent/resource | Reference or source | Identifier or catalog number |
|---|---|---|
| **Software** | | |
| PyMOL 3.1 | DeLano Scientific LLC | https://pymol.org/ |
| MEGA v7.0 | Mega Limited | https://mega.io/ |
| GraphPad Prism 10.1 | GraphPad | https://www.graphpad-prism.cn/ |
| Flowjo 10.8.1 | Treestar | https://www.flowjo.cn/ |
| Image studio | Alias | https://www.microsoft.com/ |
| ZEISS ZEN 3.12 | www.zeiss | https://www.zeiss.com/ |
| **Other** | | |
| Protein G Agarose | Beyotime | Cat #P2009 |
| HisTrapTM HP prepacked column | Cytiva | Cat # 45000513 |

## Experimental animals

Nile tilapia used in this study were purchased from an aquatic farm in Guangzhou, Guangdong Province, China. Fish were raised in an aerated circulatory system at 28 °C and fed twice a day. They had not been vaccinated and had not been exposed to the pathogens before this study. Fish with a body length of 10-12 cm were used for the experiments. Weight weeks-old female BALB/c mice were purchased and raised in the Minhang Laboratory Animal Center of East China Normal University. All experimental procedures were conducted according to the Guide for the Care and Use of Laboratory Animals of the Ministry of Science and Technology of China. Animals were randomly selected for the experiments. This study was approved by the East China Normal University Experimental Animal Ethics Committee with approval numbers of m + f20240702 and f + m20240703.

## Cell lines

Mouse A20 B-cell lymphoma cells (ATCC TIB-208) were purchased from the American Type Culture Collection (ATCC, Manassas, VA, USA). BOSC23 cells were from Wenzhou Medical University. Cells were cultured in the RPMI-1640 containing 10% FBS and 1% penicillin/streptomycin. The cells were identified and excluded the mycoplasma contamination.

## Sequence, structure, and phylogenetic analysis

The amino acid sequences were obtained from the National Center for Biotechnology Information (NCBI) GenBank. We used ExPASy (http://web.expasy.org/protparam/) and TMHMM (http://www.cbs.dtu.dk/services/TMHMM) to predict protein molecular weight and transmembrane structure. The functional domains were predicted using the Simple Modular Architecture Research Tool (SMART, http://smart.embl-heidelberg.de/). The tertiary structures of proteins were established by SWISS-MODEL (https://www.swissmodel.expasy.org/)

and shown by PyMOL. The phylogenetic trees were constructed by the neighbor-joining algorithm in MEGA v7.0 software based on multiple sequence alignment by ClustalW. Bootstrap values of 1000 replicates (%) were indicated for the branches. Information of the amino acid sequences used is listed in Appendix Table S1.

## Identification of ADPN and AdipoR genes

To identify ADPN and AdipoR genes, the Hidden Markov Model (HMM) profiles of C1q (PF00386) and HlyIII (PF03006) were downloaded from the Pfam protein family database (http://pfam.xfam.org/) and used as the query ($P < 0.01$) to search for ADPN and AdipoR genes in the genome database of the tilapia fish species. Subsequently, to further confirm the ADPN and AdipoR genes, the one-to-one orthologous genes were extracted from the genomes of multiple fish species and other vertebrates, and multiple sequence alignments were generated. In brief, the protein sequences of ADPN and AdipoR from *Danio rerio*, *Oncorhynchus mykiss*, *Clupea harengus*, *Cyprinus carpio*, *Xenopus tropicalis*, *Alligator sinensis*, *Gallus gallus*, *Homo sapiens*, and *Mus musculus* as a bit in the genomes of tilapia fish species. The query protein sequences were obtained from the Ensembl (http://useast.ensembl.org) and National Center for Biotechnology Information (NCBI) (https://www.ncbi.nlm.nih.gov/) databases. Information on the amino acid sequences used or identified is listed in Appendix Table S1.

## Bacterial infection

*Aeromonas hydrophila* was cultured in LB liquid medium at 37 °C, and were collected, washed, and resuspended in sterile PBS. Each tilapia was intraperitoneally (*i.p.*) injected with 100 μl *A. hydrophila* at the final concentration of $3.6 \times 10^6$ CFU/ml, and the control group was injected with the same volume of sterile PBS.

## Leukocyte isolation

Spleen leukocytes were isolated using Percoll density gradient centrifugation according to our previous report (Wei et al, 2019). Briefly, Percoll solution was prepared by mixing Percoll (GE Healthcare) with 10× PBS at a ratio of 9:1 and further diluted with L-15 medium (Gibco) to 52% or 34% Percoll, respectively. Then, 4 ml of 52% Percoll and 4 ml of 34% Percoll were sequentially stacked into a 15 ml tube to obtain discontinuous density layers, with cell suspension on top. After centrifugation at 500 g for 30 min with the lowest acceleration and deceleration, leukocytes were harvested from the cell layer between 52% and 34% Percoll, then washed twice, and resuspended in L-15 medium (10% FBS). We considered these isolated spleen leukocytes as lymphocytes, as over 90% of these cells belong to the lymphocyte population (Ai et al, 2022; Li et al, 2023b; Li et al, 2023c; Zhang et al, 2024b). To isolate splenocytes of mouse, the spleen cells were treated with ACK buffer to remove red blood cells, and then washed and resuspended in DMEM (10% FBS, 1% penicillin and streptomycin) for further experiments.

## Development and purification of polyclonal antibodies

The AdipoR1 antibody was generated by the Sangon Biotech Company, China. The rats were immunized for polyclonal antibody development. In the first immunization, fully emulsified mixture

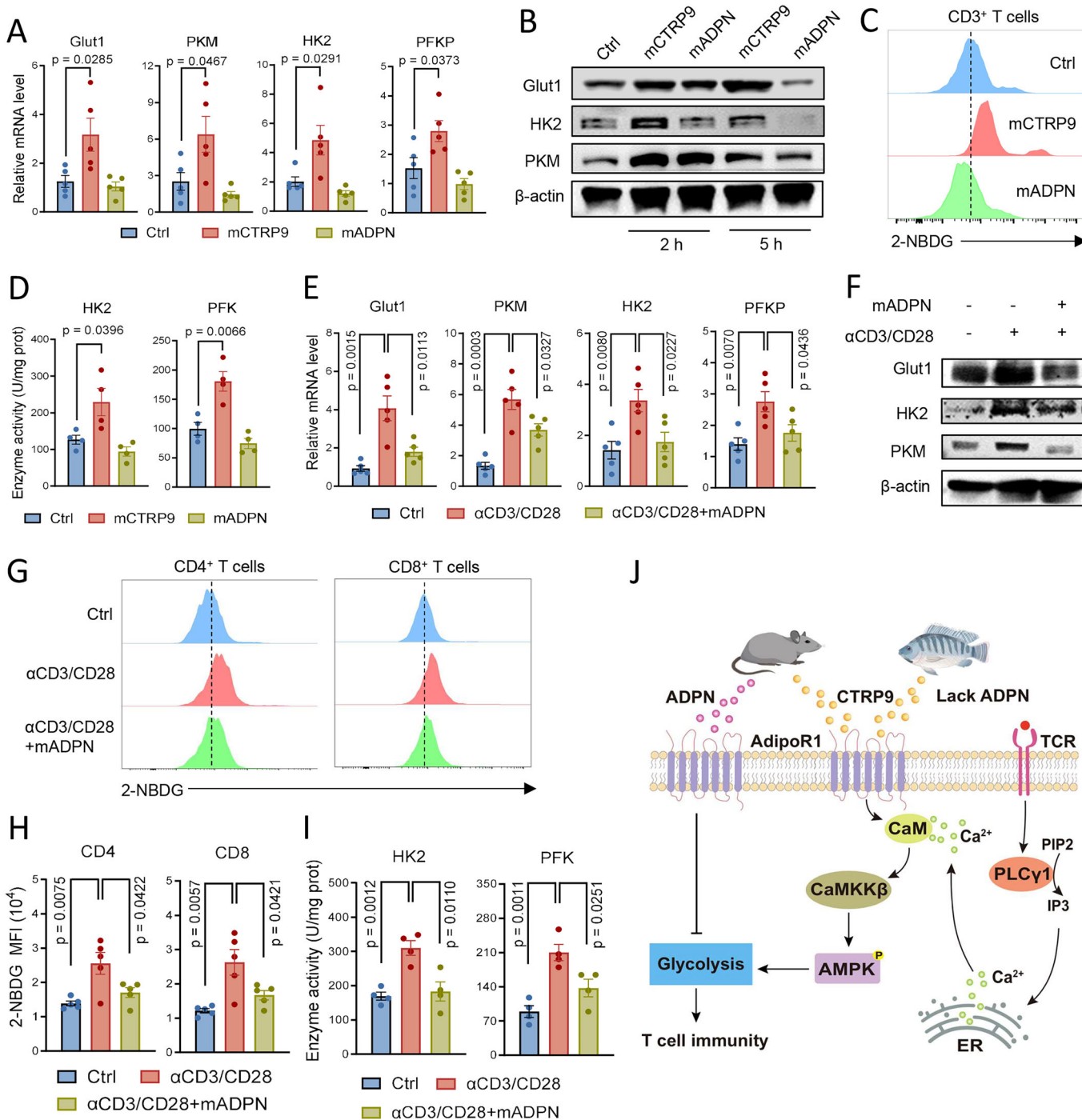

**Figure 8. CTRP9 and ADPN differentially regulate T-cell glycolysis in mouse.**

(A–D) Mouse splenocytes were stimulated with mCTRP9 or mADPN. (A) Relative mRNA levels of the indicated molecules were examined by qPCR at 5 h, $n = 5$. (B) Western blot showing protein levels of the indicated molecules at 2 h or 5 h. (C) Glucose uptake in gated CD3$^+$ T cells was measured by 2-NBDG staining at 5 h. (D) Enzyme activities of HK2 and PFK at 5 h, $n = 4$. (E–I) Mouse splenocytes were stimulated with anti-mouse CD3 plus anti-mouse CD28 in the presence of mADPN. (E) Relative mRNA levels of the indicated molecules were examined by qPCR at 5 h, $n = 5$. (F) Western blot showing protein levels of the indicated molecules at 2 h. (G, H) Glucose uptake in gated CD4$^+$ or CD8$^+$ T cells was measured by 2-NBDG staining at 5 h (G), and a bar figure (H) showing the MFI of 2-NBDG in gated CD4$^+$ and CD8$^+$ T cells, $n = 5$. (I) Enzyme activities of HK2 and PFK at 5 h, $n = 4$. (J) Schematic representation of CTRP9 and ADPN differentially regulating T-cell responses by glycolysis and crosstalk between AdipoR1 and T-cell signaling in tilapia and mouse. Data information: The $n$ stands for biological replicates. Error bars indicate mean ± SEM. Significance between the groups was determined by a two-tailed Student's $t$ test.

containing 250 µg of tilapia AdipoR1 antigenic polypeptide (NH2-C$_{362}$FRYGLEGGGCTDDSLL$_{376}$-COOH) and complete Freund's adjuvant was injected subcutaneously. Two weeks later, the second immunization was performed using the mixture of antigen with incomplete Freund's adjuvant as above. The rats were then immunized with 150 µg of the antigenic polypeptide in PBS into the tail vein for another two times with a 1-week interval. One week after the last immunization, rats were killed to obtain the serum. For antibody purification, antiserum was added to Protein G Agarose and incubated at 4 °C overnight. The beads were washed six times with pre-cooled 1×PBS, and eluted with 0.1 M Glycine-HCl (pH = 2.8). Then 1/10 volume of 1 M Tris-HCl (pH = 8.5) was added to neutralize the pH. Antibody concentration was determined using the BCA Protein Assay Kit (Sangon Biotech), and antibody specificity was detected by western blot, ELISA, and flow cytometry assays.

## ELISA assay

ELISA plate was coated with $1 \times 10^{-3}$, $1 \times 10^{-2}$, $1 \times 10^{-1}$, $1 \times 10^{0}$, $1 \times 10^{1}$, $1 \times 10^{2}$, $1 \times 10^{3}$ µg tilapia AdipoR1 antigenic polypeptide in 50 mM carbonate–bicarbonate buffer (pH 9.6) at 4 °C overnight. PBS was used as the blank control. After blocking with PBS containing 1% BSA at 37 °C for 1 h, 100-fold-diluted anti-AdipoR1 serum was added, and then incubated at 37 °C for another 1 h. After washing three times with PBST, the samples were stained with AP-conjugated goat anti-rat IgG (Solarbio, 1:4000) at 37 °C for 50 min. Subsequently, 0.1% (w/v) p-nitrophenyl phosphate (pNPP) was added and induced in the dark at 37 °C for 30 min. Finally, the reaction was terminated with 2 M NaOH, and the OD (405 nm) value was measured using a microplate reader.

## Blockage of AdipoR1 with antibody

For in vitro blockage, the isolated spleen leukocytes were cultured in glucose-free DMEM medium containing 10% FBS and 1% penicillin/streptomycin at 28 °C with 2 µg/mL of AdipoR1 antibody for the indicated time. For in vivo blockage, tilapia individuals were *i.p.* injected with 2.5 mg/kg of AdipoR1 antibody every other day during *A. hydrophila* infection or other treatment. Eventually, the cells were collected for Western blot or flow cytometry assays.

## siRNA interference

Three pairs of siRNAs targeting tilapia CTRP9, and one pair of negative control siRNA were designed and synthesized by GenePharma (Shanghai, China). Tilapia infected with *A. hydrophila* were *i.p.* injected with 3 µg CTRP9-specific siRNA or control siRNA in 100 µl 1:500 diluted Lipo6000 on days 2, 4, and 6 post infection. The primer sequences of the CTRP9 and control siRNA were listed in Appendix Table S2.

## Prokaryotic protein expression

The CDS fragments of tilapia CTRP9 and mouse CTRP9 were cloned into vector pET-28a, which were transformed into BL21(DE3) pLysS Chemically Competent Cells. The positive strain was induced by 0.5 mM IPTG at 37 °C for 4 h. The recombinant tilapia CTRP9 and mouse CTRP9 were purified by affinity chromatography using a HisTrapTM

HP prepacked column. Then, purified proteins were re-folded by gradient dialysis at 4 °C in dialysis solution with urea concentrations of 6 mmol/L, 4 mmol/L, 2 mmol/L, 1 mmol/L and 0 mmol/L (pH = 8.5), and finally dialyzed in PBS thoroughly. The purified protein was analyzed by 12% SDS-PAGE and stained with Coomassie brilliant blue. The used primers were listed in Appendix Table S2.

## Cell stimulation

For recombinant protein stimulation, $1 \times 10^{6}$ spleen leukocytes of tilapia were cultured in DMEM containing 10% FBS and 1% penicillin/streptomycin at 28 °C and stimulated with 6 µg/mL of OnCTRP9 recombinant proteins for the indicated time. In total, $1 \times 10^{6}$ mouse splenocytes were stimulated with 10 µg/mL mCTRP9 or 3 µg/mL mADPN for the indicated time. For in vivo stimulation, the tilapia individuals were *i.p.* injected with 5 mg/kg OnCTRP9 recombinant protein every 2 days during *A. hydrophilus* infection. For phytohemagglutinin (PHA, Sigma) stimulation, spleen leukocytes were cultured in DMEM containing 10% FBS and stimulated with 2 µg/mL PHA for the indicated time periods. For in vitro activation of AdipoR1, the spleen leukocytes cultured in DMEM (10% FBS, 1% penicillin/streptomycin) were treated with 10 µM AdipoRon (MedChemExpress) for the indicated time. For in vitro inhibition of AMPK, the spleen leukocytes were cultured in DMEM (10% FBS, 1% penicillin/streptomycin) containing 10 µM Compound C (MedChemExpress) for 6 h. For T-cell activation, $1 \times 10^{6}$ mouse splenocytes were stimulated with 2 µg/mL of anti-mouse CD3 (Catalog #100301, BioLegend) plus anti-mouse CD28 (Catalog #102101, BioLegend) for the indicated time; $1 \times 10^{6}$ spleen leukocytes of tilapia were stimulated with 2 µg/mL of anti-tilapia CD3 plus anti-tilapia CD28 for the indicated time points (Li et al, 2023b). For P + I stimulation, $1 \times 10^{6}$ mouse splenocytes were stimulated with 50 ng/mL phorbol 12-myrustate 13-acetate (PMA, MedChemExpress) plus 500 ng/mL ionomycin (MedChemExpress) for the indicated time. The stimulated cells were collected for Western blot and flow cytometry assays.

## Flow cytometry and cell sorting

For T-cell activation markers staining, tilapia spleen leukocytes were incubated with mouse anti-tilapia CD122 mAb on ice for 30 min (Geng et al, 2024). After washing twice with FACS buffer (PBS containing 2% FBS), the cells were further stained with 1:2000-diluted Alexa Fluor 647-conjugated goat anti-mouse IgG H&L (Catalog # ab150115, Abcam) on ice for 30 min, and followed by 1:400-diluted FITC-conjugated mouse anti-tilapia CD3ε mAb staining on ice for 30 min (Li et al, 2023b). Mouse splenocytes were incubated with 1:400 diluted FITC-conjugated anti-mouse CD4 (Catalog #100509, BioLegend), PE-conjugated anti-mouse CD8α (Catalog #162303, BioLegend), APC-conjugated anti-mouse CD69 (Catalog #104513, BioLegend), or PerCP/Cyanine 5.5-conjugated anti-mouse CD25 (Catalog #101911, BioLegend) on ice for 30 min. For AdipoR1 staining, tilapia leukocytes were first stained with 1:400 diluted FITC-conjugated mouse anti-tilapia CD3ε mAb and 4 µg/mL Rat anti-tilapia AdipoR1 polyclonal antibody on ice for 30 min. After washing twice with FACS buffer, 1:1000 diluted Alexa Fluor 647-conjugated goat anti-rat (Catalog #ab150159, Abcam) was added and incubated on ice for 30 min. The mixture of mouse splenocytes and A20 cells was stained with 1:400 diluted anti-

mouse CD4, anti-mouse CD8α, PE-Cy7-conjugated anti-mouse CD19 (Catalog #115520, BioLegend), or BV421-conjugated anti-mouse CD107a (Catalog #121617, BioLegend) on ice for 30 min. All the samples were collected by a BD LSRFortessa flow cytometer and analyzed using Flowjo software. For sorting of T cells, spleen leukocytes were stained with mouse anti-tilapia CD3ε mAb as above. After washing twice with FACS buffer, the cells were resuspended in DMEM containing 10% FBS and sorted with a BD FACSAria II flow cytometer.

## Intracellular staining

Tilapia spleen leukocytes were incubated with CD3ε mAb, followed by staining with 1:2000-diluted Alexa Fluor 647-conjugated goat anti-mouse IgG H&L on ice for 30 min. Then, cells were fixed with BD cytofix/cytoperm buffer (BD Biosciences) on ice for 30 min, and washed twice with 1× BD perm/wash buffer (BD Biosciences). Subsequently, cells were stained with 1:300 diluted mouse anti-tilapia IL-2 or mouse anti-tilapia granzyme B (Cao et al, 2024; Zhang et al, 2024a), cleaved caspase 3 (Catalog #9664, Cell Signaling Technology), cleaved caspase 8 (Catalog #9496, Cell Signaling Technology), p-AMPKα antibody (Thr172, Catalog #2535, Cell Signaling Technology), CaM (Catalog #4830, Cell Signaling Technology) or CaMKKβ (Catalog #16810, Cell Signaling Technology) on ice for 30 min (Li et al, 2023c). After staining with anti-mouse CD4, CD8α, and PE-Cy7-conjugated anti-mouse CD19 as above, the mixture of mouse splenocytes and A20 cells was fixed with BD cytofix/cytoperm buffer on ice for 30 min. Subsequently, cells were stained with cleaved caspase 3, cleaved caspase 8, APC-conjugated anti-mouse IL-2 (Catalog #503809, BioLegend), PE-conjugated anti-mouse TNF-α (Catalog #506305, BioLegend), or Brilliant Violet 421-conjugated anti-mouse IFN-γ (Catalog #505829, BioLegend), PE anti-human/mouse Granzyme B Recombinant (Catalog #372207, BioLegend) on ice for 30 min. For p-AMPKα (Thr172), CaM and CaMKKβ staining, tilapia spleen leukocytes stained with FITC-conjugated mouse anti-tilapia CD3ε as above, followed by fixation with BD cytofix/cytoperm buffer. Then, cells were incubated with 1:400-diluted p-AMPKα (Thr172), CaM or CaMKKβ mAb (Cell Signaling Technology) on ice for 30 min, after washing twice with 1× BD perm/wash buffer, 1:2000-diluted Alexa Fluor 647-conjugated goat anti-rabbit IgG H&L was added and incubated on ice for 30 min.

## Ca²⁺ influx assay

For $Ca^{2+}$ influx assay, $1 \times 10^7$ leukocytes were resuspended with 1 mL loading buffer (1% FBS and 10 mM HEPES in HBSS without phenol red), and incubated with 2.5 µg/ml Indo-1 (Life Technologies) at 30 °C for 30 min in the dark. After incubation, cells were washed twice with L-15 medium and resuspended in 500 µl loading buffer. The samples were collected using a BD LSR Fortessa flow cytometer to obtain the baseline fluorescence of the 450:510 nm ratio on the lymphocyte population. Then, 2.5 µg/mL AdipoRon or 2 µg/mL Thapsigargin was added at 120 s to induce $Ca^{2+}$ influx, and the ratios of 450:510 nm, for both baseline and after stimulation, were used for analysis.

## Glucose uptake assay

Freshly isolated spleen leukocytes were washed with pre-cooled PBS, and $1 \times 10^6$ leukocytes were resuspended with 100 µl PBS and added into a 96-well cell culture plate. The cells of tilapia or mouse were incubated with 100 mM fluorescent deoxyglucose derivative (2-NBDG; Life Technologies) at 28 °C or 37 °C for 30 min. Then, the cells were washed twice with pre-cooled PBS containing 2% FBS. Finally, the samples were analyzed by flow cytometer for glucose uptake.

## Extracellular acidification rate (ECAR) detection

In all, $1 \times 10^6$ spleen leukocytes of tilapia were cultured in DMEM containing 10% FBS and 1% penicillin/streptomycin at 28 °C and stimulated with 6 µg/mL of OnCTRP9 recombinant proteins for 3 h. After collecting the treated cells, detection of cellular acidification rate is carried out in accordance with the operating instructions of a commercial kit (Elabscience, E-BC-F069). The fluorescence microplate reader was used to detect the fluorescence intensity of each example at an excitation wavelength of 490 nm and an emission wavelength of 535 nm, and the detection was performed every 2.5 min for 120 min. Finally, the fluorescence values were recorded to calculate ECAR.

## Examination of enzyme activity

The mouse splenocytes were crushed by ultrasonication on ice. The samples were then centrifuged at 8000 rpm, 4 °C for 10 min, and the supernatant was harvested for measurement. Activities of hexokinase 2 (HK2) and phosphofructokinase (PFK) were measured with commercial assay kits (Beijing Solarbio Science & Technology Co., Ltd, China) according to the instructions of the manufacturer. The enzyme activity was calculated as per mg of protein.

## Death and apoptosis assay

Freshly isolated leukocytes were stained with CD3ε as above. Then, cells were stained with 1:400 diluted Alexa Fluor 647-conjugated Annexin V antibody (Catalog #640920, BioLegend) in Annexin V Binding buffer (0.01 M HEPES/NaOH, 0.14 M NaCl, 2.5 mM $CaCl_2$) for 15 min at room temperature (Li et al, 2023b). 1:400 diluted 7-AAD (Catalog #A1310, Invitrogen) was added to the samples shortly before collection by flow cytometer.

## BrdU incorporation

Control or *A. hydrophila*-infected tilapia was *i.p.* injected with 0.75 mg of BrdU (Catalog #19-160, Sigma) in 200 µl of PBS one day before the animals were sacrificed, and spleen leukocytes were isolated for assay. Spleen leukocytes were first stained for surface CD3ε as above. Cells were fixed with BD Cytofix/Cytoperm Buffer on ice for 30 min and washed twice with BD Perm/Wash Buffer. After treating with BD Cytoperm Plus Buffer (BD Bioscience) for 10 min and BD Cytofix/Cytoperm Buffer on ice for another 5 min, cells were then treated with 300 µg/ml DNase (Sigma) at 37 °C for 1 h and washed with BD Perm/Wash Buffer. Subsequently, samples were stained with 1:100 diluted APC-conjugated anti-BrdU antibody (Catalog #17-5071-42, Invitrogen) at room temperature for 20 min. The samples were washed twice and analyzed by flow cytometer.

## In vitro proliferation assay

In all, $1 \times 10^6$ spleen leukocytes were labeled with 10 μM CFSE (Catalog #C34554, Invitrogen) at room temperature for 9 min according to the manufacturer's protocol. After washing twice with DMEM, the labeled cells were cultured in DMEM with or without 10 μg/mL recombinant mouse CTRP9, and 2 μg/mL of anti-mouse CD3 plus 2 μg/mL of plate-bound anti-mouse CD28 were used to induce T-cell proliferation. Cells were cultured at 28℃ for 48 h and then stained with 1:400 diluted FITC-conjugated anti-mouse CD4 mAb and 1:400 diluted PE-conjugated anti-mouse CD8 mAb as above, and 1:400 diluted 7-AAD was added to identify live/dead cells shortly before flow cytometry assay.

## Immunofluorescence assay

To detect the expression of AdipoR1, the spleen leukocytes were centrifuged onto slides using Cytospin 4 (Thermo Scientific) at $1000 \times g$ for 3 min and fixed with methyl alcohol for 5 min. After blocking with 1% BSA for 1 h at room temperature, the samples were stained using anti-AdipoR1 antibody as the primary antibody and Alexa Fluor 488-conjugated goat anti-rat IgG H&L (Abcam, 1:800) as the secondary antibody. All incubations were performed at room temperature for 1 h, followed by three washes with PBS after each incubation. The cells were stained with Hoechst 33342 (Beyotime) and observed with a Zeiss ApoTome microscope.

## Construction of the CAR targeting murine CD19

A Woodchuck hepatitis virus posttranscriptional regulatory element (WPRE) was inserted behind the transgene cassette in the MSCV retroviral backbone (Clonetech). The anti-murine CD19 CAR construct, which contains a single-chain variable fragment derived from monoclonal antibody 1D3 (Kochenderfer et al, 2010), a portion of the murine 4-1BB sequence, and the cytoplasmic region of murine CD3ζ, was synthesized by Genscript. Anti-murine CD19 CAR and GFP were separated by a P2A linker and cloned into the MSCV plasmid. The vector harboring GFP served as a reporter control.

## Generation of anti-CD19 CAR-T cells

BOSC23 cells were co-transfected with 10 μg of mCD19-GFP-CAR plasmid or GFP-CAR plasmid and 5 μg of PCL-ECO helper plasmid, and the cell supernatant enriched with retroviral particles was collected at 72 h post-transfection. In total, $2 \times 10^6$ cells of mouse splenocytes were resuspended with RPMI 1640 medium in 24-well plates and treated with 2 μg/ml anti-CD3 plus 2 μg/ml anti-CD28 antibodies, and 100 U/ml IL-2. The activated cells were transferred into a six-well plate at 24 h later, and 1 ml of the retrovirus particle-rich cell supernatant with additional polybrene was added to the cells, centrifuged at 2000 rpm for 60 min, and incubated at 37 ℃ for 8 h. Then, the cells were replaced with fresh medium containing 100 U/ml IL-2. Cells were collected at 72 h after virus infection, and the frequency of GFP+CAR-T cells was determined by flow cytometry.

## Co-culture of A20 B-cell lymphoma cells with anti-CD19 CAR-T cells

The anti-CD19 CAR-T cells were incubated with A20 B-cell lymphoma cells in 24-well plates at a ratio of 1:1 ($5 \times 10^5$ A20 cells in each well) for the indicated time. CD19+ A20 cells were gated for apoptosis and cleaved caspase-3 FACS staining, and the GFP+CD4+ or GFP+CD8+ CAR-T cells were gated for IL-2, IFN-γ, TNF-α, Granzyme B, and CD107a FACS staining. Cytotoxicity of CAR-T cells was assessed as LDH release, using the Cytotoxicity Detection Kit Plus (Sigma) according to the manufacturer's instructions.

## Real-time quantitative PCR (qPCR)

Total RNA was extracted from spleen leukocytes with TRIzol reagent (Invitrogen) according to the manufacturer's instruction. After being treated with gDNA pure to remove genomic DNA, first-strand cDNA was synthesized using the NovoScript Plus All-in-one 1st Strand cDNA Synthesis SuperMix (Novoprotein). Then, the qPCR was performed with NovoStart SYBR qPCR SuperMix Plus (Novoprotein) on the QuantStudio 5 Real-Time PCR Instrument (Applied Biosystems). β-actin was selected as the internal reference and the relative gene expression levels were analyzed with the $2^{-\Delta\Delta Ct}$ method. The gene-specific primers used were listed in Appendix Table S2.

## Western blot assay

Spleen leukocytes were lysed with NP40 lysis buffer (1% NP-40, 150 mM NaCl, 50 mM Tris-HCl, pH = 7.4) containing 1 mM PMSF and 0.1% protease inhibitor on ice for 30 min. After centrifugation at 12,000 rpm, 4 ℃ for 10 min, the supernatant was collected and boiled in 5× SDS loading buffer for 5 min, then the samples were subjected to 12% SDS-PAGE. The separated proteins were transferred onto nitrocellulose (NC) membrane and blocked in PBST (PBS with 0.05% Tween-20) containing 4% non-fat powdered milk at room temperature for 1 h. After that, the blots were incubated with 1:1000 diluted primary antibodies against phospho-AMPKα (Thr172, Catalog #2535), phospho-S6 (Ser240/244, Catalog #5364), phospho-ERK1/2 (Thr202/Tyr204, Catalog #4370), phospho-AKT (Thr308, Catalog #13038), AMPKα (Catalog #5831), CaM (Catalog #4830), CaMKKβ (Catalog #16810), β-actin (Catalog #4970) from Cell Signaling Technology, phospho-JNK (Thr183/Thr183/Thr221, Catalog #AF1762), phospho-p65 (Ser536, Catalog #AF5881), Glut1 (Catalog #AF1015), HK2 (Catalog #AG2142), PKM (Catalog #AF7764), AdipoR1 (Catalog #AF2131) from Beyotime or rat anti-tilapia AdipoR1 at 4℃ overnight (Ai et al, 2022; Li et al, 2023b; Li et al, 2023c; Liang et al, 2025; Wei et al, 2019; Wei et al, 2020a). After washing three times with PBST, blots were incubated with 1:30000 diluted Alexa Fluor 800-conjugated goat-anti-rabbit IgG H&L (Catalog #5151, Cell Signaling Technology) or Alexa Fluor 680-conjugated goat-anti-rat or mouse IgG H&L (Catalog #ab175775, Abcam) as secondary antibody at room temperature for 1 h. Then, blots were washed with PBST for three times, and scanned by Odyssey CLx Image Studio.

## Statistical analysis

All data are presented as the mean ± SEM, and two-tailed Student $t$ test is applied to define the statistical significance. The $P$ values are defined as follows: *$P < 0.05$; **$P < 0.01$; ***$P < 0.001$.

# Data availability

The source data of this paper are collected in the following database: https://doi.org/10.6084/m9.figshare.30436783.

The source data of this paper are collected in the following database record: biostudies:S-SCDT-10_1038-S44319-025-00640-0.

## Peer review information

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

## Acknowledgements

We thank the Instrument-sharing Platform of the School of Life Sciences of East China Normal University (ECNU) for instrument sharing, and the Flow Cytometry Core Facility of the School of Life Sciences of ECNU for the FACS analysis. This study was supported by grants from the National Natural Science

Foundation of China (No. 32373165), the National Key Research and Development Program of China (No. 2022YFD2400804), the Natural Science Foundation of Shanghai (24ZR1419700), and Shanghai Sailing Program (24YF2710300).

## Author contributions

**Kunming Li**: Validation; Investigation; Writing—original draft. **Jiansong Zhang**: Validation; Investigation. **Kang Li**: Funding acquisition; Validation; Investigation. **Haokai Chen**: Validation; Investigation. **Wenhai Deng**: Investigation. **Wenzhuo Rao**: Writing—original draft. **Ming Geng**: Investigation. **Yuying Zheng**: Investigation. **Xiumei Wei**: Supervision; Funding acquisition; Validation; Project administration; Writing—review and editing. **Jialong Yang**: Supervision; Funding acquisition; Validation; Methodology; Writing—original draft; Project administration; Writing—review and editing.

Source data underlying figure panels in this paper may have individual authorship assigned. Where available, figure panel/source data authorship is listed in the following database record: biostudies:S-SCDT-10_1038-S44319-025-00640-0.

## Disclosure and competing interests statement

The authors declare no competing interests.

