## [Peer Review File · EMBO Reports]

CTRP9 enhances T cell immunity by binding AdipoR1 that cross-talks with TCR and promotes glycolysis

Kunming Li, Jiansong Zhang, Kang Li, Haokai Chen, Wenhai Deng, Wenzhuo Rao, Ming Geng, Yuying Zheng, Xiumei Wei, and Jialong Yang

Corresponding author(s): Jialong Yang (jlyang@bio.ecnu.edu.cn) , Xiumei Wei (xmwei@bio.ecnu.edu.cn)

Review Timeline:

Submission Date:	19th Mar 25
Editorial Decision:	12th May 25
Revision Received:	8th Aug 25
Editorial Decision:	26th Sep 25
Revision Received:	1st Oct 25
Accepted:	3rd Nov 25

Editor: Achim Breiling

Transaction Report:

Dear Prof. Yang,

Thank you for the submission of your manuscript to EMBO reports. I have now received the reports from the three referees that were asked to evaluate your study, which can be found at the end of this email.

As you will see, the referees think that these findings are of interest. However, they have several comments, concerns, and suggestions, indicating that a major revision of the manuscript is necessary to allow publication of the study in EMBO reports. As the reports are below, and all the referee concerns need to be addressed, I will not detail them here.

Given the constructive referee comments, I would like to invite you to revise your manuscript with the understanding that the concerns of the referees must be addressed in the revised manuscript and/or in a detailed point-by-point response. Acceptance of your manuscript will depend on a positive outcome of a second round of review. It is EMBO reports policy to allow a single round of revision only and acceptance of the manuscript will therefore depend on the completeness of your responses included in the next, final version of the manuscript.

1) a .docx formatted version of the final manuscript text (including legends for main figures, EV figures and tables), but without the figures included. Figure legends should be compiled at the end of the manuscript text.

2) individual production quality figure files as .eps, .tif, .jpg (one file per figure), of main figures and EV figures. Please upload these as separate, individual files upon re-submission.

4) a complete author checklist, which you can download from our author guidelines

(<https://www.embopress.org/page/journal/14693178/authorguide>). Please insert page numbers in the checklist to indicate where the requested information can be found in the manuscript. The completed author checklist will also be part of the RPF.

5) that primary datasets produced in this study (e.g. RNA-seq, ChIP-seq, structural and array data) are deposited in an

appropriate public database. If no primary datasets have been deposited, please also state this in a dedicated section (e.g. 'No primary datasets have been generated and deposited'), see below.

The accession numbers and database should be listed in a formal "Data Availability" section that follows the model below. This is now mandatory (like the COI statement). Please note that the Data Availability Section is restricted to new primary data that are part of this study. This section is mandatory. As indicated above, if no primary datasets have been deposited, please state this in this section

Data availability

6) We now request the publication of original source data with the aim of making primary data more accessible and transparent to the reader. You will receive a separate email with instructions for providing source data with your revised manuscript, including information how to upload and organize the files.

8) Regarding data quantification and statistics, please make sure that the number "n" for how many independent experiments were performed, their nature (biological versus technical replicates), the bars and error bars (e.g. SEM, SD) and the test used to calculate p-values is indicated in the respective figure legends (also for EV and Appendix figures). Please also check that all the p-values are explained in the legend, and that these fit to those shown in the figure. Please provide statistical testing where applicable. Please avoid the phrase 'independent experiment', but clearly state if these were biological or technical replicates. Please also indicate (e.g. with n.s.) if testing was performed, but the differences are not significant. In case n=2, please show the data as separate datapoints without error bars and statistics. See also: <http://www.embopress.org/page/journal/14693178/authorguide#statisticalanalysis>

9) Please add scale bars of similar style and thickness to microscopic images, using clearly visible black or white bars (depending on the background). Please place these in the lower right corner of the images themselves. Please do not write on or near the bars in the image but define the size in the respective figure legend.

10) Please also note our reference format:

12) We now use CRedit to specify the contributions of each author in the journal submission system. CRedit replaces the author contribution section. Please use the free text box to provide more detailed descriptions and do NOT provide your final manuscript text file with an author contributions section. See also our guide to authors: <https://www.embopress.org/page/journal/14693178/authorguide#authorshippinguidelines>

13) All Materials and Methods need to be described in the main text using our 'Structured Methods' format, which is required for

all research articles. According to this format, the Methods section should include a Reagents and Tools Table (listing key reagents, experimental models, software, and relevant equipment and including their sources and relevant identifiers), uploaded as separate file, and a Methods section in which we encourage the authors to describe their methods using a step-by-step protocol format with bullet points, to facilitate the adoption of the methodologies across labs. More information on how to adhere to this format as well as downloadable templates (.doc) for the Reagents and Tools Table can be found in our author guidelines (section 'Structured Methods'):

14) Please add up to five keywords to the manuscript and order the sections like this, using these names:
Title page - Abstract - Keywords - Introduction - Results - Discussion - Methods - Data availability section - Acknowledgements - Disclosure and Competing Interests Statement - References - Figure legends - Expanded View Figure legends

15) Please make sure that all the funding information is also entered into the online submission system and that it is complete and similar to the one in the acknowledgement section of the manuscript text file.

I look forward to seeing a revised form of your manuscript when it is ready.

Please use this link to submit your revision: <https://embor.msubmit.net/cgi-bin/main.plex>

Yours sincerely,

Referee #1:

In the current study, the authors show in Tilapia and mouse models that CTRP-9 suppresses enhances T cell glycolysis, T cell effector functions and anti-microbial defense. Further, in mouse models a role for this CTRP-9- AdipoR1 axis in tumour T cell cytotoxicity is shown. While the findings are interesting, the manuscript should address the following issues

1. Data in Figure 2 and 3 demonstrate the role of AdipoR1 in anti-bacterial immunity and T cell effector mechanisms. Here, it is not clear if CTRP-9 binding to AdipoR1 is required for these effects. This should be shown in further invitro and invivo experiments
2. Data in Figure 4 shows that invitro treatment of spleen leukocytes with rCTRP-9 enhances Ca²⁺-CaMKK β -AMPK axis. This data should be further confirmed with r CTRP-9 treatment in vivo as well.
3. During infection with *A. hydrophila*, the authors should demonstrate the increased expression of CTRP-9 in Tilapia.
4. What is the effect of glycolysis inhibition on the CTRP-9-AdipoR1 axis in Tilapia leukocytes. This should be demonstrated by means of invitro experiments
5. How does the CTRP-9-AdipoR1 axis control glycolysis in the Tilapia model? Is it via AMPK?. Also, sea-horse assays should be done to confirm increased extra-cellular acidification rates in the presence of rCTRP-9
6. While in the Tilapia model, a role for CTRP-9-AdipoR1 in anti-bacterial immune defense is shown, the study shows a role for the same axis in tumor mediated cytotoxicity in the mouse model. It would be interesting the assess if the anti-bacterial immunity mediated by CTRP-9-AdipoR1 axis is seen in the mouse model as well.
7. Figure 9A and B, please show the expression of AdipoR1 and AdipoR2 in these experimental set-ups

The manuscript can be accepted after the issues raised have been addressed

Referee #2:

The manuscript from Kunming Li et al utilizes mouse and tilapia models to explore the function of two AdipoR ligands, ADPN and CTRP9. Tilapia have AdipoR receptor and CTRP9, but lack ADPN, enabling the study of CTRP9-AdipoR interactions in the absence of ADPN. AdipoR1 is upregulated following infection with *A. hydrophila* in tilapia. In tilapia, CTRP9 or AdipoR treatment results in changes to CaMKKB pathway protein abundance or phosphorylation. Treatment with CAMKK-inhibitor STO-609 results in diminished T cell activity. Mouse CD4⁺ T cells increased IL-2 production and IFN γ production when cultured in the presence of CTRP9. In an in vitro setting, mouse anti-CD19 CAR T cells cultured with mCTR9 resulted in an increase in apoptotic A20 target cells. Glycolysis-related mRNA and proteins are upregulated following mCTR9 treatment and resulted in increased glucose uptake. Taken together, these data suggest that CTRP9 promotes T cell glycolysis in tilapia and mice.

This work has several important shortcomings in experimental design and/or data interpretation:

- Some presented data is unnecessary or redundant and - most critically - inconsistent. The inconsistency raises concerns about the reliability of these data and/or representation. A key example is Figure 2. In Figure 2C-D, approx. 40% of lymphocytes are AdipoR1⁺ at 5dpi, with a tight spread in replicates. In contrast, Figure 2F-G shows approx. 26% of lymphocytes as AdipoR1⁺ at 5 dpi. Figure 2F-G also conflicts with the subsequent panels in Figure 2. For example, Figure 2F-G show approximately 4% of CD3⁻ cells are AdipoR1⁺ ($3.24/(3.24+69.5)$), while this value is 20.7% in Figure 2M. Perhaps this issue is due to improperly labeled Figure 2F-G, which could be showing the uninfected population at "5 dpi", but nonetheless requires clarification. The authors may also be better served by showing changes in total numbers of AdipoR1⁺ T cells rather than changes in percent CD3⁺ or AdipoR1⁺.
- Most experiments were performed on unpurified splenocytes, so the adipokines could be affecting T cells in an indirect manner. As such, how can the authors exclude that AdipoR1 ligands may be acting on non-T cells, which exert an effect on T cells? Figure 2M-N show an increase in AdipoR1⁺ cells among the CD3⁻ cells during infection, which may be involved.
- Related to Figure 3, the mechanism by which anti-AdipoR1 is resulting in more fish death and T cell dysfunction during infection is unclear. Is this effect due solely to blocking AdipoR1/ligand binding? Or does antibody-binding to this surface protein result in opsonization or otherwise block expansion/trigger apoptosis? Complementary experiments may be informative: e.g. sequestering CTRP9 or titrating in CTRP9 to examine effects on AdipoR1⁺ cells.
- Please clarify the design and interpretation of Figure 1H. As currently described, it seems like the authors co-transfected the same cells with constructs for ligands and receptors. If this is the case, then interpretation of their presence within the same cell does not indicate ligand-receptor interaction as described, but rather represents their co-transfection.
- Please clarify the experimental design in Figure 8. E.g. what conditions allow the Ctrl and mCTR9-treated values to be paired in Figure 8Q-R? Are they from T cells from the same donor mouse? Technical replicates? If significance values were calculated without paired datapoints, there would likely not be a significant difference between the Ctrl and mCTR9-treated conditions for CD4⁺ T cells.
- Why are mCTR9's effects mostly confined to CD4⁺ T cells (Figure 7)? Do CD4⁺ cells upregulate AdipoR1 more than CD8⁺ T cells in response to infection? Or is it a difference in downstream signaling pathways? Are there Treg-specific effects?
- In the discussion, please describe or speculate how the two ligands binding the same receptor cause opposite effects. Do they result in different conformational changes? Or is it, as referenced above, due to effects of AdipoR1-ligands on non-T cell splenocytes that in turn signal to T cells?
- Please explicitly clarify in the text and abstract that the CAR-T killing result is in an in vitro killing assay. Please also clarify that the tertiary structures presented are predicted structures, rather than solved structures (e.g. Fig 1A, 1F).
- In the discussion, it is written "We found that the majority of T cells in tilapia expressed AdipoR1, and most AdipoR1⁺ were indeed T cells..." This investigation was limited to the spleen, so the authors cannot say that most AdipoR1⁺ cells in Tilapia are T cells; there are likely other abundant AdipoR1⁺ populations outside of spleen.
- Minor point: most of the figures include representative flow cytometry plots plus summary bar plots, which are not all necessary to include as main figures. Most representative flow plots can reasonably be included as supplementary materials instead. These data could more succinctly be conveyed, in fewer than 9 main figures.

Referee #3:

This manuscript explores the functional relevance of CTRP9-AdipoR1 signaling in T cell activation and metabolism, using both *Oreochromis niloticus* (tilapia) and murine models. The experimental rationale is sound and original, especially considering the evolutionary context in which tilapia has lost the gene encoding adiponectin (ADPN), retaining only CTRP9 as a functional

AdipoR1 ligand. This natural divergence represents an elegant model to dissect the ligand-specific effects on AdipoR1 signaling. The authors convincingly demonstrate that CTRP9 binding to AdipoR1 promotes T cell activation via calcium influx and the CaM-CaMKK β -AMPK pathway, ultimately enhancing glycolysis and antimicrobial responses. Similar findings are extended to the murine system, including improved CAR-T cell efficacy. Overall, the study provides novel insights into adipokine-mediated immunometabolic regulation of T cells.

However, several key issues need to be addressed before the manuscript can be considered for publication.

Major points:

1. Validation of anti-AdipoR1 antibody in tilapia

The specificity of the anti-AdipoR1 antibody used in flow cytometry and neutralization assays is unclear. Polyclonal antibodies generated against tilapia AdipoR1 might recognize other targets. This is particularly relevant, since it is not monospecific. A preadsorption control using purified antigen should be included to validate specificity.

2. Figs 6H-I: Glucose depletion model

The rationale for using a complete glucose-deprivation condition needs clarification. Total absence of glucose is a harsh and physiologically unlikely condition. Testing T cell apoptosis and function under low-glucose conditions would be more relevant and mechanistically informative. Furthermore, the finding that recombinant CTRP9 rescues apoptosis in the absence of glucose requires mechanistic explanation.

3. Antibody specificity in tilapia

Several figures rely on commercial antibodies for detecting signaling molecules or surface markers in tilapia. These antibodies must be validated for cross-reactivity with tilapia proteins. At the very least, catalog numbers and relevant validation references (if available) should be included in the Materials and Methods.

4. In vivo infection model: role of T cells

The in vivo bacterial infection model is central to the conclusions regarding CTRP9-mediated T cell protection. However, the manuscript does not justify the relevance of T cells in primary antibacterial responses in fish. In teleosts, innate responses dominate early defense against bacterial pathogens. Were the fish vaccinated or previously exposed to the pathogen to generate antigen-specific T cells? This should be clarified to support the claim that CTRP9 enhances T cell-mediated protection.

Minor points:

1. Line 582: Please replace "prokaryotic protein recombination" with "prokaryotic protein expression."

2. Line 629: The correct term is "intracellular staining," not "intercellular staining."

3. The glucose transporter "GLUT1" is misspelled multiple times in both text and figures.

4. Fig. 11: The two lanes shown in the immunoblot are not labeled. Please specify the experimental conditions each lane represents.

5. Please, use correct gene and protein symbols nomenclature for tilapia.

Point-by-point Responses to reviewers' comments

We gratefully appreciate the editors and reviewer for their time and insightful comments, which have enabled us to improve the quality of our work. During the revision, each comment has been fully considered, and our responses and revisions are indicated as below point by point.

Referee #1:

In the current study, the authors show in Tilapia and mouse models that CTRP-9 enhances T cell glycolysis, T cell effector functions and anti-microbial defense. Further, in mouse models a role for this CTRP-9-AdipoR1 axis in tumor T cell cytotoxicity is shown. While the findings are interesting, the manuscript should address the following issues.

Response: We would like to thank the reviewer for thinking our study interesting.

1. Data in Figure 2 and 3 demonstrate the role of AdipoR1 in anti-bacterial immunity and T cell effector mechanisms. Here, it is not clear if CTRP-9 binding to AdipoR1 is required for these effects. This should be shown in further *in vitro* and *in vivo* experiments.

Response: We appreciate this suggestion. As advised, we knocked down CTRP9 using siRNA during bacterial infection. We found that interference of CTRP9 impaired the activation of AdipoR1-CaMKK β -AMPK signaling and AdipoR1⁺ cells expansion, resulted in a defect of T cell activation, survival and function, thereby impairing the pathogen clearance and causing a higher mortality of tilapia. These new data are shown in **Figure 4F-4R**, **Figure S8B-S8G**, confirming CTRP9-AdipoR1 signaling is crucial for the proper T cell response.

2. Data in Figure 4 shows that *in vitro* treatment of spleen leukocytes with rCTRP-9 enhances Ca²⁺-CaMKK β -AMPK axis. This data should be further confirmed with rCTRP-9 treatment *in vivo* as well.

Response: As advised, we treated tilapia with recombinant CTRP-9 *in vivo*, and then examined the activation of Ca²⁺-CaMKK β -AMPK signaling pathway at both mRNA and protein levels. The new data are showed in **Figure 4E**, **Figure S5F**.

3. During infection with *A. hydrophila*, the authors should demonstrate the increased expression of CTRP-9 in Tilapia.

Response: Because the antibody for tilapia CTRP9 is not available, we examined the expression of CTRP9 during *A. hydrophila* infection at mRNA level by qPCR. The results showed that *A. hydrophila* infection induced the expression of CTRP9 in spleen leukocytes. The new data are showed in **Figure S8A**.

4. What is the effect of glycolysis inhibition on the CTRP-9-AdipoR1 axis in Tilapia leukocytes. This should be demonstrated by means of *in vitro* experiments.

Response: We appreciate this comment. As advised, we treated spleen leukocytes of tilapia using the glycolytic inhibitor 2-Deoxy-D-glucose (MCE, HY-13966) *in vitro*, and examined the expression of CTRP-9-AdipoR1 signaling components. As below, we found glycolysis inhibition had no effect on the transcription of CTRP9, CaM, CaMKK β , and AMPK α 1, but slightly reduced the mRNA level of AdipoR1. At protein levels, glycolysis inhibition did not affect CaM and AMPK α expression, but slightly reduced the CaMKK β expression and AMPK α phosphorylation. These results suggest glycolysis may potentially regulate CTRP-9-AdipoR1 signaling via a feedback regulation while CTRP-9-AdipoR1 axis prompting the glycolysis. However, considering this is not the key evidence for this regulation, we decide not to present these data in the Figure.

5. How does the CTRP-9-AdipoR1 axis control glycolysis in the Tilapia model? Is it via AMPK? Also, sea-horse assays should be done to confirm increased extra-cellular acidification rates in the presence of rCTR-9.

Response: To investigate whether CTRP9-AdipoR1 axis control glycolysis via AMPK, spleen leukocytes of tilapia were treated with rCTR-9 in the presence or absence of AMPK inhibitor Compound C. The results showed that inhibiting AMPK activity impaired the CTRP9-induced upregulation of HK2, PKM and PFKF,

indicating CTRP9 may prompt glycolysis via AMPK signaling. The new data are showed in **Figure 5R**.

For extracellular acidification rates (ECAR) assay, because Seahorse analyzer is not available in our university, we used a commercial detection kit (Elabscience, E-BC-F069), which is also used in published studies for instead ^[1, 2]. We found that rCTRP9 treatment increased the ECAR, and the new data are showed in **Figure 5N**.

[1] α -Lipoic Acid Ameliorates Arsenic-Induced Lipid Disorders by Promoting Peroxisomal β -Oxidation and Reducing Lipophagy in Chicken Hepatocyte. *Adv Sci.* 2025, 12(11): e2413255.

[2] A subcellular selective APEX2-based proximity labeling used for identifying mitochondrial G-quadruplex DNA binding proteins. *Nucleic Acids Res.* 2025, 53(1): 259.

6. While in the Tilapia model, a role for CTRP-9-AdipoR1 in anti-bacterial immune defense is shown, the study shows a role for the same axis in tumor mediated cytotoxicity in the mouse model. It would be interesting the assess if the anti-bacterial immunity mediated by CTRP-9-AdipoR1 axis is seen in the mouse model as well.

Response: We appreciate this valuable suggestion, and we strongly agree with the reviewer. However, we are sorry to say that Laboratory Animal Center of East China Normal University is not certified to conduct bacterial or viral infections in mice, so we are not permitted to perform any pathogen infection experiment in mice. We therefore hope the Editor and reviewer would allow us to investigate this issue in the future. Thanks.

7. Figure 9A and B, please show the expression of AdipoR1 and AdipoR2 in these experimental set-ups.

Response: As below, we examined the expression of AdipoR1 and AdipoR2, except the AdipoR2 protein level because its antibody is not available. The new data regarding AdipoR1 are showed in **Figure S12A** and **S12B**. Because we did not mention AdipoR2 throughout the manuscript, we decide not to present the AdipoR2 data in the Figure.

Referee #2:

The manuscript from Kunming Li et al utilizes mouse and tilapia models to explore the function of two AdipoR ligands, ADPN and CTRP9. Tilapia have AdipoR receptor and CTRP9, but lack ADPN, enabling the study of CTRP9-AdipoR interactions in the absence of ADPN. AdipoR1 is upregulated following infection with *A. hydrophila* in tilapia. In tilapia, CTRP9 or AdipoRon treatment results in changes to CaMKKB pathway protein abundance or phosphorylation. Treatment with CAMKK-inhibitor STO-609 results in diminished T cell activity. Mouse CD4⁺ T cells increased IL-2 production and IFN γ production when cultured in the presence of CTRP9. In an *in vitro* setting, mouse anti-CD19 CAR T cells cultured with mCTRP9 resulted in an increase in apoptotic A20 target cells. Glycolysis-related mRNA and proteins are upregulated following mCTRP9 treatment and resulted in increased glucose uptake. Taken together, these data suggest that CTRP9 promotes T cell glycolysis in tilapia and mice. This work has several important shortcomings in experimental design and/or data interpretation:

1. Some presented data is unnecessary or redundant and - most critically - inconsistent. The inconsistency raises concerns about the reliability of these data and/or representation. A key example is Figure 2. In Figure 2C-D, approx. 40% of lymphocytes are AdipoR1⁺ at 5dpi, with a tight spread in replicates. In contrast, Figure 2F-G shows approx. 26% of lymphocytes as AdipoR1⁺ at 5 dpi. Figure 2F-G also conflicts with the subsequent panels in Figure 2. For example, Figure 2F-G show approximately 4% of CD3⁻ cells are AdipoR1⁺ ($3.24/(3.24+69.5)$), while this value is 20.7% in Figure 2M. Perhaps this issue is due to improperly labeled Figure 2F-G, which could be showing the uninfected population at "5 dpi", but nonetheless requires clarification. The authors may also be better served by showing changes in total

numbers of AdipoR1⁺ T cells rather than changes in percent CD3⁺ or AdipoR1⁺.

Response: We appreciate the reviewer for pointing out this issue. We apologize for the confusion. As the reviewer speculated, Figures 2F and 2G are indeed uninfected samples, which we mistakenly labeled as “5 dpi” in the Figure legend. We have corrected this mistake. In addition, because we aim to investigate the association between CD3⁺ and AdipoR1⁺ cells, we think that the cell percentage would be more appropriate than the cell number.

2. Most experiments were performed on unpurified splenocytes, so the adipokines could be affecting T cells in an indirect manner. As such, how can the authors exclude that AdipoR1 ligands may be acting on non-T cells, which exert an effect on T cells? Figure 2M-N show an increase in AdipoR1⁺ cells among the CD3⁻ cells during infection, which may be involved.

Response: We appreciate the reviewer for raising this issue. To exclude effect of non-T cells, we sorted CD3⁺ T cells from the spleen of healthy tilapia and treated them with recombinant CTRP9 *in vitro* in the presence or absence of CD3/CD28 mAbs. Then, we examined the expression level of IL-2, CD122, IFN- γ , CD44, Perforin A, Granzyme B, Glut1, PKM, HK2, CaM, CaMKK β , AMPK α 1, AMPK α 2 and AMPK γ 1, to confirm the direct effects of CTRP9 on CaM–CaMKK β –AMPK signaling in T cells, and the direct regulation of CTRP9 on T cell activation, function and glycolysis. These new data are shown in **Figure 5E, 5F, 5K, S5G**. These results can help to exclude the effects of non-T cells at some extent.

3. Related to Figure 3, the mechanism by which anti-AdipoR1 is resulting in more fish death and T cell dysfunction during infection is unclear. Is this effect due solely to blocking AdipoR1/ligand binding? Or does antibody-binding to this surface protein result in opsonization or otherwise block expansion/trigger apoptosis? Complementary experiments may be informative: e.g. sequestering CTRP9 or titrating in CTRP9 to examine effects on AdipoR1⁺ cells.

Response: We appreciate the reviewer for pointing out this issue. As advised, we knocked down the CTRP9 using siRNA during bacterial infection, and found that interference of CTRP9 impaired the expansion of AdipoR1⁺ cell, resulting in a defect of T cell activation, survival and function. These new data are shown in **Figure 4F-4R**,

Figure S8B-S8G, confirming CTRP9-AdipoR1 signaling is crucial for the proper T cell response.

4. Please clarify the design and interpretation of Figure 1H. As currently described, it seems like the authors co-transfected the same cells with constructs for ligands and receptors. If this is the case, then interpretation of their presence within the same cell does not indicate ligand-receptor interaction as described, but rather represents their co-transfection.

Response: We agree with the reviewer that this result cannot indicate ligand-receptor interaction. Considering we already have the FACS data (Figure 1J in the revised version) indicating the interaction between CTRP9 and AdipoR1, we decide to remove this co-transfection data. Thanks.

5. Please clarify the experimental design in Figure 8. E.g. what conditions allow the Ctrl and mCTRP9-treated values to be paired in Figure 8Q-R? Are they from T cells from the same donor mouse? Technical replicates? If significance values were calculated without paired datapoints, there would likely not be a significant difference between the Ctrl and mCTRP9-treated conditions for CD4⁺ T cells.

Response: Yes, all CAR-T data in this Figure have used pair-way to analyze. We generated CAR-T cells from the same donor mouse, and then treated the CAR-T cells with mCTRP9 or not. These two samples (treated with mCTRP9 or not) from the same mouse were considered as one pair. Four-pair means we generated CAR-T cells from four mice. So they are biological replicates, but not technical replicates. As advised, we have clarified this in the Figure legends. Another reason why we used pair-way is that we cannot finish all the experiments of four mice in one time, because there are too many experiments. We completed all these four pairs by two or three times, thereby pair-way is appropriate to use in this case to avoid differences between different batches of experiments. We hope the Editor and reviewer would agree with us.

6. Why are mCTRP9's effects mostly confined to CD4⁺ T cells (Figure 7)? Do CD4⁺ cells upregulate AdipoR1 more than CD8⁺ T cells in response to infection? Or is it a difference in downstream signaling pathways? Are there Treg-specific effects?

Response: We appreciate this comment. As advised, we added some new experiments.

We found that CD4⁺ cells upregulate AdipoR1 similarly with CD8⁺ T cells upon TCR stimulation. Meanwhile mCTRP9 addition did not change the percentage of CD4⁺CD25⁺ Treg cells during CD3/CD28 mAb stimulation (see below).

However, during P+I stimulation, the presence of mCTRP9 induced higher levels of T-bet and NF-κB in CD4⁺ T cells, which are the important transcription factors for IFN-γ and IL-2, respectively. In contrast, the CTRP9-induced upregulation of transcription factors cannot be observed in CD8⁺ T cells. These results suggest that CTRP9 differently regulates the cytokine production of CD4⁺ and CD8⁺ T cells, because it induces different expression patterns of the key transcription factors in these two populations. The new data are showed in **Figure 6L-6O**.

7. In the discussion, please describe or speculate how the two ligands binding the same receptor cause opposite effects. Do they result in different conformational changes? Or is it, as referenced above, due to effects of AdipoR1-ligands on non-T cell splenocytes that in turn signal to T cells?

Response: We added some discussion about this issue in the revised manuscript, where we speculate that the binding of ADPN and CTRP9 to the same receptor causing different effects may be a result of different conformational changes, and this was also reported in TNFR and GPCR [3, 4].

[3] Molecular determinants of ligand efficacy and potency in GPCR signaling. *Science*. 2023, 382: eadh1859

[4] Disabling TNF receptor signaling by induced conformational perturbation of tryptophan-107. *Proc Natl Acad Sci U S A*. 2005, 102: 10970-10975

8. Please explicitly clarify in the text and abstract that the CAR-T killing result is in an in vitro killing assay. Please also clarify that the tertiary structures presented are predicted structures, rather than solved structures (e.g. Fig 1A, 1F).

Response: As advised, we have clarified the CAR-T killing result is an *in vitro* killing

assay in Results and Abstract, and clarified the tertiary structures presented are predicted structures in the Figure legend.

9. In the discussion, it is written "We found that the majority of T cells in tilapia expressed AdipoR1, and most AdipoR1⁺ were indeed T cells..." This investigation was limited to the spleen, so the authors cannot say that most AdipoR1⁺ cells in Tilapia are T cells; there are likely other abundant AdipoR1⁺ populations outside of spleen.

Response: We appreciate this comment. As suggested, we have limited this conclusion to the spleen.

10. Minor point: most of the figures include representative flow cytometry plots plus summary bar plots, which are not all necessary to include as main figures. Most representative flow plots can reasonably be included as supplementary materials instead. These data could more succinctly be conveyed, in fewer than 9 main figures.

Response: We appreciate the reviewer for raising this issue. As advised, we have moved some data to the supplementary materials. Now we have 8 main figures.

Referee #3:

This manuscript explores the functional relevance of CTRP9-AdipoR1 signaling in T cell activation and metabolism, using both *Oreochromis niloticus* (tilapia) and murine models. The experimental rationale is sound and original, especially considering the evolutionary context in which tilapia has lost the gene encoding adiponectin (ADPN), retaining only CTRP9 as a functional AdipoR1 ligand. This natural divergence represents an elegant model to dissect the ligand-specific effects on AdipoR1 signaling. The authors convincingly demonstrate that CTRP9 binding to AdipoR1 promotes T cell activation via calcium influx and the CaM-CaMKK β -AMPK pathway, ultimately enhancing glycolysis and antimicrobial responses. Similar findings are extended to the murine system, including improved CAR-T cell efficacy. Overall, the study provides novel insights into adipokine-mediated immunometabolic regulation of T cells. However, several key issues need to be addressed before the manuscript can be considered for publication.

Response: We would like to thank the reviewer for confirming the originality and

novelty of our study.

Major points:

1. Validation of anti-AdipoR1 antibody in tilapia. The specificity of the anti-AdipoR1 antibody used in flow cytometry and neutralization assays is unclear. Polyclonal antibodies generated against tilapia AdipoR1 might recognize other targets. This is particularly relevant, since it is not monospecific. A preadsorption control using purified antigen should be included to validate specificity.

Response: We appreciate the reviewer for raising this issue. Actually, we validated the AdipoR1 antibody by western blot in our original manuscript, it is showed in **Figure 1H**. In which, we used this antibody to probe the protein from tilapia spleen leukocytes, and observed a single band matching the molecular weight of tilapia AdipoR1, suggesting its specificity.

In addition, during this round of revision, we also used ELISA assay to validate the specific recognition between antigen and this antibody. The new data are showed in **Figure S2E**.

2. Figs 6H-I: Glucose depletion model. The rationale for using a complete glucose-deprivation condition needs clarification. Total absence of glucose is a harsh and physiologically unlikely condition. Testing T cell apoptosis and function under low-glucose conditions would be more relevant and mechanistically informative. Furthermore, the finding that recombinant CTRP9 rescues apoptosis in the absence of glucose requires mechanistic explanation.

Response: We appreciate the reviewer for raising this issue. Actually, complete deprivation of glucose is a common used approach ^[5-6]. But as advised, we used a low level of glucose (25% of the normal level) to instead no-glucose setting. Consistent with results observed upon complete glucose deprivation, low-glucose conditions also increased apoptosis in tilapia T cells, and the presence of rCTRP9 rescued low-glucose-induced apoptosis. The new data were showed in **Figure 5G and 5H**.

For the mechanism, we found CTRP9 alleviated low-glucose-induced T cell apoptosis by reducing the cleavage of Caspase-3 and Caspase-8. The new data were showed in **Figure 5I, 5J, and Figure S9D, S9E**.

[5] The energy sensor AMPK regulates T cell metabolic adaptation and effector responses in vivo. *Immunity*, 2015, 42(1), 41–54.

[6] Energy sensor AMPK gamma regulates translation via phosphatase PPP6C independent of AMPK alpha. *Molecular cell*, 2022, 82(24), 4700–4711.e12.

3. Antibody specificity in tilapia. Several figures rely on commercial antibodies for detecting signaling molecules or surface markers in tilapia. These antibodies must be validated for cross-reactivity with tilapia proteins. At the very least, catalog numbers and relevant validation references (if available) should be included in the Materials and Methods.

Response: In this manuscript, antibodies for CaM, CaMKK β , AMPK, Glut1, HK2, PKM, Annexin-V, cleaved caspase-3, cleaved caspase-8, phospho-NF- κ B p65, phospho-ERK1/2 and phospho-AMPK were used to detect tilapia proteins. Except for CaMKK β antibody, other antibodies have been used in tilapia in our previous reports [7-12], and we have cited these references in the manuscript. As advised, we have added the catalog numbers of all the used antibodies in the Materials and Methods.

We also validated the cross-reactivity of these antibodies with tilapia proteins. As shown by the uncropped western blots image (see below), these antibodies could specifically recognize the corresponding proteins in tilapia.

In addition, we found the amino acid sequences of the antigen peptides using for antibody generation are conserved (see below). Therefore, the cross-reactive of this antibody with tilapia is reasonable.

antigen for p-NFκB p65 antibody: peptides around Ser276 of human NFκB p65

human NFκB p65	S	M	Q	L	R	R	P	S	D	R	E	L	S	E	P	M	E	F	Q	Y	L	P
Nile tilapia NFκB p65	K	M	Q	L	R	R	P	S	D	R	E	V	S	E	P	M	D	F	Q	Y	L	P

antigen for p-ERK1/2 (Thr202/Tyr204) antibody: peptides around Thr202/Tyr204 of human p44 MAPK

human	DFGLARI	ADPEHDHTGFL	TEYVATR	WYRAPEI	MLNS	KGY
Tilapia	DFGLARI	ADPEHDHTGFL	TEYVATR	WYRAPEI	MLNS	KGY
Consensus	dfglari	adpehdhtgfl	teyvatr	wyrapei	mlns	kgy

antigen for p-AMPKα (Thr172) antibody: peptides around Thr172 of human AMPKα2

Human	MMSDGEFLRTS	CGSPNYAAPEV
Tilapia	MMSDGEFLRTS	CGSPNYAAPEV
Consensus	mmsdgeflrts	cgs pnyapev

antigen for AMPKα (Arg21) antibody: peptides around Arg21 of human AMPKα2

Human	VAVKI	LNRQKIRSLD	VVGKIRREI	QNLKLF	RHPHI	IK
Tilapia	VAVKI	LNRQKIRSLD	VVGKIRREI	QNLKLF	RHPHI	IK
Consensus	vavki	lnrqkirs	ldvvgkrr	ei qnlklfr	hphi	ik

antigen for human Glut1 antibody

human	MEPSSK KKL TGR LMLAVGGAVLGS LQF GYNTGVI NAPQKVI EEFYNQTWVHRYGESI LPTITL TLWSLS VAI FSVG
tilapia	MD. QGRQI TLPL LMS VGTAVI GSLQF GYNTGVI NAPQRI I ENFI NQTWSYRYNEPI SKISLTAVMSI AVAI FSVG
Consensus	m k t l vg av gslqfgyntgvnapqkie f nqtw ry e i t lt ws vaifsvg
human	GM GSFS VGLFVNRF GRRNS MLMNLLAFVS AVLMGFS KLGKS FEMLI LGRELI I GVCYGLT TGFVPMYVGEVSP T
tilapia	GI FGSFS VGLFVNRL GRRNS MLANI LAFI S AVLMGFS KMAKS WEMLI I GRFVVGLYCGLS TGFVPMYVGEI SPT
Consensus	g gsfsvglfvnr grrnsm m n laf savlmgfsk ks emli grf g ycglt tgfvp yvge spt
human	ALRGALGTLHQLGI VVGI LI AQVFLGDSI VGNKDLWPLLLSII FI PALLQCI VLPFCPES PRFLLI NRNEENRAK
tilapia	ALRGALGTLHQLGI VI GI LI AQI FGLEAI VGNNDLWPLLLAFLEI PAVI QCVL LPLCPES PRFLLI NKNEENKAK
Consensus	alrgalgtlhqlgivi giliaq fgl imgn lwp lll fipa qc lp cpesprfl lin neen ak
human	SVLKKLRGTADV THDLQENKEES RQMMREKKVTI LELFRS PAYRQPI LI AVVLQLS QQLS GI NAVFYYS TSI FEK
tilapia	SVLKKLRGTADV SADMQENKEES RQMMREKKVTI LELFRS PLYRQPLI AVMLQLS QQLS GI NAVFYYS TSI FEK
Consensus	svlklrgt dv d qemkeesrqmmrekkvtilelfrsp yrqp li av lqlsqqls gi navfyystsifek
human	AGVQQPVYATI GS GI VNTAFVTVS LFVVERAGRRT LHLI GLAGMAGCAI LMTI ALALLEQLP WMS YLSI VAI FGF
tilapia	AGVEQPI YATI GAGVNTAFVTVS LFVVERAGRRLHL LGLMGMAGS AI LMTI ALALLDQLR WMS YVSI VAI FAF
Consensus	agv qp yatig g vntaftvsvlfvveragr r lhl gl gmag ailmtialall ql wms y sivaif f
human	VAFFEVGP GPI PWFI VAELFS QGPRPAI AVAGFS NWT S NFI VGMCFQYVEQL CGPYVFI I FTVLLV LFFI FTYF
tilapia	VAFFEI GP GPI PWFI VAELXS QGPRPS AI AVAGFS NWT ANFI VGMGFQYV ADACGPYVFI I FTVLLV I FFI FTYF
Consensus	vaffe gpgpi pwfivael sqgprp aiavagfswt nfivgmfqyv cgpyvf iftvllv fffityf
human	KVPETKGRTFDEI ASGFRQGGAS QS DKTPEELFHP LGADS Q
tilapia	KVPETKGRTFDDI TAGFRQTS AAEKHTPEELNS. LGADS Q
Consensus	kvp etkgrtfd i gfrq a tpeel lgadsq

antigen for human Annexin V antibody

human	MAQVLRGT V TDFPGF DERADAETLRKAMKGLGTDEESI L TLLTS RS NAQRQE I SAAFKTFLGRDLDDDLKSELTG
tilapia	. . MAYRGS VRPYNE NAKHDAEI LHKAMKGI GTDEDAI LMLLTARS NDQRQI KAAYKKAHGKDLV SALKSELGG
Consensus	rg v f dae l kamkg gtde il llt rsn qrqi aa k g dl lksel g
human	KFEKLI VALNKPS RLYDAYE I KHALKGAGTNEKVLTEI I ASRTPEELRAI KQVVEEYESS LEDVVGDTSGYYQ
tilapia	LFESLI VALNTPS VLYDATLH NALKGAGTEDE VLI EI LASRTGEQI KEI TKYKKEFGGKLEKDI CGDTS GHYQ
Consensus	fe livalmpsl yda l alkgagt vl ei asrte i vy e g le d gdtsg yq
human	RMLVLLQANRDPDAGI DEAQVEQDAQALFQAGELKWTGDEEKFI TI FGTRSVS HLRKVE DKYMTI SFGQI EETI
tilapia	KLVI LLQGS RE. . EGVDDEKI EKDAKDLYAAGEEF GTDEEKFI TI LGNRS AEHLRKYFAAYKLSGS DI EDSI
Consensus	lv llq r g de e da l age k gtdeekf it i g rs hlrkvf y sg ie i
human	DRETS GNLEQLLAVVKS I RSI PAYLAETLYYANKGAGTDDHTLI RVVRSRSEI DLFNI RKEFRKNFATSLYSM
tilapia	KGETTGNLENLLLAVVKAESI PNFFAERLYKSMRRAGTDDDTLMRI VVRSRSEV DMLDI RASFKNVYGQSLYTTI
Consensus	et gnle lllavvk sip ae ly m agtdd tlr mvsrse d ir f k sly i
human	KGDTSGDYKKALLLLCGED
tilapia	QEDTGDYQKALLYLCCGN
Consensus	dt gdy kall leg

antigen for human PKM antibody

human	AFI QTQQLHAANADTFLEHVCRLDI DS PPI TARNTGI I CTI GPASRSVETLKEM KS GNVVARLNFS HGTHEYHAE TI KN
tilapia	AFI QTQQLYAATADTFLEHVCLLDI DS FPI TARNTGI I CTI GPASRSYDI LKEM KS GNI ARLNFS HGTHEYHAQTI KN
Consensus	afiqtql aa adtflehmc l dids pitarntgiictigpasrsv lkem ksgnm arlnfshgttheyha tikn
human	VRTATESFASDPI LYRPVAV ALDTKGPEI RTGLI KGS GTA EVELKKGATLKI TLDNAYVEKCDENI LWLDYKNI CKVVEV
tilapia	YREACESFEPGSI QYRPI GI ALDTKGPEI RTGLI KGS GTA EVELKKGNM KI TLDSDYQENCSEDI LWLDYKNI TKVVEV
Consensus	vr a esf i yrp aldtkgpeirtgliksgtaevelkkg kitld y e c e ilwldykni kvvev
human	GSKI YVDDGLI SLQVKQKGFVTEVENGSGS LGS KRGVNLPGA AVDLP A
tilapia	GSKVYI DDGLI SLQVKEI GADFLNCEI ENGGT LGS KRGVNLPGA AVDLP A
Consensus	gsk y ddglislqvkgadfl e eng lsgskgvnlpgaavdlpa

antigen for human CaM antibody

human	NQADQLTEEQI AEFKEAFS LFDKDGDTI TTKELGTVMRS LGQNPTEAELQDM NEVDADGNGTI DFPEFLTMA
tilapia	. MADQLTEEQI AEFKEAFS LFDKDGDTI TTKELGTVMRS LGQNPTEAELQDM NEVDADGNGTI DFPEFLTMA
Consensus	adqlteeqi aefkeafslfdkdgdgtittkelgtvmslgqnptaelqdm nevdadngnti dfpeflmma
human	RKMKDTDS EEEI REAFRVFDKDGNGYI SAAELRHVMTNLGEKLTDEEVDEMI READI DGDGQVNYEEFVQMMA
tilapia	RKMKDTDS EEEI REAFRVFDKDGNGYI SAAELRHVMTNLGEKLTDEEVDEMI READI DGDGQVNYEEFVQMMA
Consensus	rkmkdt dseeci reafrvfdkdgngyisaaelrhvmtnlgekl tdeevdemi readi dgdgqvnyee f vqmnta

antigen for mouse HK2 antibody

mouse	MIASHMIA CLFTELNQNQVQKVDQYL YHMRLS DETLLEI SRRFRKEMEKGLGATTHPTAAVKMLPTFVRS TPDGTEHGGEF
tilapia	NSTSNPLANYS TELHDDQADKVDKYLHNLQLS DKTLMVDSLRFREMDKGLCRDINPTAAVKMLPTFVRS TPDGTEQGGEF
Consensus	m s a t e l q k v d y l l s d t l s r f r e m k g l t p t a a v k m l p t f v r s t p d g t e g e f
mouse	LALDLGGT NFRVL RVRVTDNGLQRVENENQI YAI PEDI MRGS GTQLFDHI AECLANFMDKLOI KEKKLPLGFTFS FPCQHQ
tilapia	LALDLGGS NFRVLLVKVMNGEQKVEVES QI YDI PEHI MRGS GSEFFDHI ADCLANYLDKVGMKDKKLPGLFTFS FPCQOQ
Consensus	l a l d l g g n f r v l r v r v t d n g l q r v e n e n q i y a i p e d i m r g s g t q l f d h i a e c l a n f m d k l o i k e k k l p l g f t f s f p c q h q
mouse	TKLDES..... FLVSWTKGFKSSGVEGRDVEDLI RKAIRRRGDFDI DI VAVVNDTVGTMTTCGYDDQN
tilapia	TKLDEI TPPQPNIQPSLSLNPQAVLNSWTKGFRSSGVEGHVDVSLLRKSI KKRGRDFDI DI VAVVNDTVGTMTTCGFDDRH
Consensus	t k l d e s f l v s w t k g f k s s g v e g r d v e d l i r k a i r r r g d f d i d i v a v v n d t v g t m t t c g y d d q n
mouse	CEI GLI VGTGS NACYMEEMRHI DMVEGDEGRVCI NMEWGAFGDDGTINDI RTEFDREI DMVCSLNPQKQLFEKMI SGNVYG
tilapia	CEVGLI VGTGTNACYMEQMRNI GVLDGDEGRMVCNTEWGAFGDDGALDRLTDI DRELDAGSFNPGKQLFEKMI SGNVYG
Consensus	c e g l i v g t g n a c y m e e m r h i d m v e g d e g r v c i n m e w g a f g d d g t i n d i r t e f d r e i d m v c s l n p q k q l f e k m i s g n v y g
mouse	ELVRLI LVKMAKAE LFFQGLSPELLTTGFS FETKDVSDI EDDKD. . GI QKAYQI LVRGLSFLQEDCVATHRI CQI VSTR
tilapia	ELVRLI LVKVARSQLLFQGTTS ELLTTGHFS TSHI YAI ENDKDEEGI ASAEKI LRS LGLDPSVEDCI ATRRVCQI VSTR
Consensus	e l v r l i l v k m a k a e l f f q g l s p e l l t t g f s f e t k d v s d i e d d k d . . g i q k a y q i l v r g l s f l q e d c v a t h r i c q i v s t r
mouse	SASLCAATLA AVLWRI KENKGEERLRS TI GVDGS VYKHPHF AKRLHKA VRRRLVPDCDVRFLRS EDGS GKGAAMTAVAY
tilapia	AAHLCASSLAS VLRQI RDNKAAEKLRVTI GVDGS VYKNHPEFS RRLNKVRRRLVPDCDVRFLQS QDGS GKGAAMTAVAF
Consensus	s a s l c a a t l a a v l w r i k e n k g e e r l r s t i g v d g s v y k h p h f a k r l h k a v r r r l v p d c d v r f l r s e d g s g k g a a m t a v a y
mouse	RLADQHRARQKTLES I KLS HEOLLEV KRRMKVEME QGLS KETHEAAPV KMLPTYV CATPDGTEK GDFLALDLGGT NFRVL
tilapia	RLANQNAERQRI LDTLRSL REOLLEV KRRFS EEMTRGLS KQTHQOASI KMLPTYVRS TPDGS EHGDFLALDLGGS SFRVL
Consensus	r l a d q h r a r q k t l e s i k l s h e o l l e v k r r m k v e m e q g l s k e t h e a a p v k m l p t y v c a t p d g t e k g d f l a l d l g g t n f r v l
mouse	LVRVRNGKRRGVEV HNKI YSI PQEVV HGTGEELFDHI VQCI ADFLEYVGMKGVSLPLGFTFS FPCQONS LDQSI LLKWKI
tilapia	LVRVRS GTKRS VDYQOKI YSI PQEIMOGTGEELFNHI VDCMADELEYVGMKGVSLPLGFTFS FPCDQTKLDEGI LLKWKI
Consensus	l v r v r n g k r r g v e v h n k i y s i p q e v v h g t g e e l f d h i v q c i a d f l e y v g m k g v s l p l g f t f s f p c q o n s l d q s i l l k w k i
mouse	GFKAS GCEGEDVVTLLKEAI RRREFDL DVAVVNDTVGTMTTCGYEDPHCEVGLI VGTGS NACYMEEMRNVVELVDGEEG
tilapia	GFKAS GCEGKDYVALLKEAVRS RGEFDLNFVAVVNDTVGTMTTCGYEDPKCEVGLI VGTGTNACYMEEMHNVVELVDGNG
Consensus	g f k a s g c e g e d v v t l l k e a i r r r e f d l d v a v v n d t v g t m t t c g y e d p h c e v g l i v g t g s n a c y m e e m r n v e l v d g e e g
mouse	RVCVNMEWGAFGDNGCI DDLRIVFDVAVDLS LNP GKQRF EKMI SGNYLGEI VRNI LI DFTKRGLLFRGRI SERLKTRGI
tilapia	RVCNVMEWGAFGENGLEEFCTEFDRLYDACS NYP GKQRYEKMI SGNYLGEI VRNVLLDFTAKGLLFRGKVS SERLKTRGI
Consensus	r m e v n m e w g a f g d n g c i d d l r i v f d v a v d l s l n p g k q r f e k m i s g n y l g e i v r n i l i d f t k r g l l f r g r i s e r l k t r g i
mouse	FETKFLS QI ES DCLALLQVRAI LRHLGLE. STCDDSI I VKEVCTVVARAAQLCGAGMAAVVDKI RENRGLDNLKVTVGV
tilapia	FETKFLS QI ES DRLAVQVRS I LOHLGLTGS TCDDSVL VKEVCS VVARAAQLS GAGLAAVVDKI RONRNLNQLSI TVGV
Consensus	f e t k f l s q i e s d c l a l l q v r a i l r h l g l e . s t c d d s i i v k e v c t v v a r r a a q l c g a g m a a v v d k i r e n r g l d n l k v t v g v
mouse	DGTLYKLHPHF AKVWHETVRDLAPKCDVSVFLS EDGS GKGAALI TAVACRI REAGQ
tilapia	DGTLYKTHPHFS AI MQETLRDLAPQCEVTF LKS EDGS GKGAALI TAVACRVENEQQ
Consensus	d g t l y k l h p h f a k v w h e t v r d l a p k c d v s v f l s e d g s g k g a a l i t a v a c r i r e a g q

antigen for CaMKKβ: Peptides around Gly228 of human CaMKKβ protein

Human CaMKKβ	RQAGFPRRPPPRGTRP APGCI QPRGPI EQVYQEI AI LKKLDHPNVVKLVEVLDDP
Tilapia CaMKKβ	RQAGFPRRPPPRGARA APEGPAQPKGPLERVYQEI AI LKKLDHPNVVKLVEVLDDP
Consensus	r q a g f p r r p p p r g r a p g q p g e v y q e i a i l k k l d h p n v v k l v e v l d d p

antigen for Cleaved Caspase-3 antibody: peptides around Asp175 of human Caspase-3

Human	I QACRGTELD CGI ETDS GVDDDMACHKI PVEADFLY
Tilapia	I QACRGTDLD PGI ETDS ETDG. . . VKI PVEANFLY
Consensus	i q a c r g t l d g i e t d s d k i p v e a f l y

antigen for Cleaved Caspase-8: Peptides around Asp387 of human Caspase-8 protein

human	LAGKPKVFFI QACQGDNYQK
tilapia	LGGKPKLFFI QACQGEYQK
Consensus	l g k p k f f i q a c q g y q k

[7] Glutamine Metabolism Underlies the Functional Similarity of T Cells between Nile Tilapia and Tetrapod. *Adv Sci.* 2023, 10(12): e2201164.

[8] Dietary restriction to optimize T cell immunity is an ancient survival strategy conserved in

vertebrate evolution, *Cell Mol Life Sci*, 2023, 80(8): 219.

[9] IL-2-mTORC1 signaling coordinates the STAT1/T-bet axis to ensure Th1 cell differentiation and anti-bacterial immune response in fish. *PLoS Pathog*. 2022, 18(10):e1010913.

[10] Ancestral T Cells in Fish Require mTORC1-Coupled Immune Signals and Metabolic Programming for Proper Activation and Function. *J Immunol*. 2019, 203(5): 1172-1188.

[11] Ca²⁺-Calcineurin Axis-Controlled NFAT Nuclear Translocation Is Crucial for Optimal T Cell Immunity in an Early Vertebrate. *J Immunol*. 2020, 204(3): 569-585.

[12] Dual phosphorylation of glycogen synthase kinase 3 β differentially integrates metabolic programs to determine T cell immunity across vertebrates. *Cell Mol Life Sci*. 2025, 28;82(1):218.

4. *In vivo* infection model: role of T cells. The *in vivo* bacterial infection model is central to the conclusions regarding CTRP9-mediated T cell protection. However, the manuscript does not justify the relevance of T cells in primary antibacterial responses in fish. In teleosts, innate responses dominate early defense against bacterial pathogens. Were the fish vaccinated or previously exposed to the pathogen to generate antigen-specific T cells? This should be clarified to support the claim that CTRP9 enhances T cell-mediated protection.

Response: We appreciate the reviewer for raising this issue. We have investigated the relevance of T cells with primary antibacterial responses in fish in our previous studies [7, 9, 13]. In these studies, we found that depletion of CD3⁺ T cells, CD4-1⁺ T cells or CD8⁺ T cells impaired the anti-bacterial immune response of tilapia. In addition, the tilapia used in this study were purchased from a certified hatchery. They had not been vaccinated, and had not exposed to the pathogens before this study. As advised, we have added this information in the section of “Materials and Methods”.

[7] Glutamine Metabolism Underlies the Functional Similarity of T Cells between Nile Tilapia and Tetrapod. *Adv Sci*. 2023, 10(12): e2201164.

[9] IL-2-mTORC1 signaling coordinates the STAT1/T-bet axis to ensure Th1 cell differentiation and anti-bacterial immune response in fish. *PLoS Pathog*. 2022, 18(10):e1010913.

[13] Fish requires FasL to facilitate CD8⁺ T-cell function and antimicrobial immunity. *J Immunol*. 2025, 214(6):1219-1235.

Minor points:

1. Line 582: Please replace "prokaryotic protein recombination" with "prokaryotic protein expression."
2. Line 629: The correct term is "intracellular staining," not "intercellular staining."
3. The glucose transporter "GLUT1" is misspelled multiple times in both text and figures.
4. Fig. 1I: The two lanes shown in the immunoblot are not labeled. Please specify the experimental conditions each lane represents.
5. Please, use correct gene and protein symbols nomenclature for tilapia.

Response: We appreciate the reviewer for pointing out these issues. As advised, we have made corresponding corrections in the revised manuscript.

Dear Prof. Yang

Thank you for the submission of your revised manuscript to EMBO reports. Since my colleague Achim Breiling is currently out of office, I have temporarily taken over the handling of your manuscript. As I already informed you, we have meanwhile received the full set of referee reports that is copied below. All three referees support publication of your study in EMBO reports.

Browsing through the manuscript myself, I noticed a few editorial things that we need before we can proceed with the official acceptance of your study.

- On the abstract page of the manuscript, please include 4-5 general keyword terms to enhance searchability.
- We requested the ORCID ID number being linked to the account in our online system for Dr Xiumei Wei, as this is obligatory for all co-corresponding authors. This information is currently missing.
- Regarding the Author Contributions, we now use CRediT to specify the contributions of each author in the journal submission system. Therefore, please remove the Author Contributions from the manuscript file and make sure that the author contributions in our online manuscript tracking system are correct and up-to-date. The information you specified in the system will be automatically retrieved and typeset into the article. You can enter additional information in the free text box provided, if you wish.
- The Data availability section should only refer to data deposited in public repositories, such as the source data you list. The sentence "All data needed to evaluate the conclusions in the paper are present in the paper or the Supplementary Materials" should be removed.
- Please rename the file with supplementary figures and tables "Appendix". Please add a table of contents with page numbers and correct the nomenclature to "Appendix Figure S1" etc. and "Appendix Table S1" etc. throughout. Please also correct the citations in the manuscript text.
- Appendix Figure S2E: please specify 'n' in the legend and whether it is biological or technical replicates.
- Appendix Fig. S3, S5, S6, S7, S8, S9, S12: please specify the exact p-values, define the bars (mean, median) and error bars, and the nature of the replicates 'n' (biological or technical).
- Reagent table: please correct the citation of Table S2, which should be Appendix Table S2.
- Materials and Methods should be Methods
- During our routine image checks, we noticed that the images across the figure set appear pixelated under analysis. This is a common result of converting original 16-bit TIFF images to RGB format for publication, and while not a cause for concern, it can sometimes give the impression of image alteration to critical readers.

To resolve this please upload the figure set at a higher resolution.

- Our production/data editors have asked you to clarify several points in the figure legends (see below). Please incorporate these changes in the manuscript and return the revised file with tracked changes with your final manuscript submission.

A) Statistical test information. Only p-values that are actually shown in the figure panel(s) should (and must) be defined in the legends, all others should be removed from (or added to) the legend. Moreover, we ask for the specification of exact p-values, unless the p-values are small ($p < 0.0001$), which can be reported as inequalities:

1. Please note that the exact p values are not provided in the legends of figures 2A, D, N; 3B, D, G, I, K, L, M; 4B, G, J, K, M, O, P, Q, R; 5B, D, E, F, H, J, K, N, P; 6D, G, H, J, K, M, O; 7E, G, I, K, M, O, Q, R; 8A, D, E, H, I.

B) Replicates and error bars:

2. Please note that the error bars are not defined in the legends of figures 2A, D, G, I, K, N; 3B, D, G, I, K, L, M; 4B, G, J, K, M, O, P, Q; 5B, D, E, F, K, N, P; 6D, 8A, D, E, H, I.

- As a standard procedure we edit the title and abstract to make it more accessible to our general readership. Please find my suggestions below my signature.

- Finally, EMBO Reports papers are accompanied online by

A) a short (1-2 sentences) summary of the findings and their significance,

B) 2-3 bullet points highlighting key results and

C) a schematic summary figure that provides a sketch of the major findings (not a data image).

Please provide the summary figure as a separate file in PNG or JPG format at a size of 550x300-600 pixels (width x height).

Please note that the size is rather small and that text needs to be readable at the final size. Please send us this information along with the revised manuscript.

With kind regards,

=====

Referee #1:

The authors have revised the manuscript according to the comments raised and the manuscript can now be accepted for publication

Referee #2:

The authors' revisions and additional experiments have improved the manuscript. Thank you for addressing the reviewer feedback.

Referee #3:

The authors have satisfactorily addressed all the concerns raised in my previous review. The revised manuscript is now clear, scientifically sound, and of potential interest to the readership of EMBO Reports. I therefore recommend acceptance in its current form.

=====

CTRP9 engages AdipoR1 and promotes T cell glycolysis and immunity

The adiponectin (ADPN) receptor (AdipoR) modulates T-cell responses, but its effects remain controversial since signaling can either promote or inhibit T cell function. Interaction with the ligand ADPN inhibits T-cell responses but given the existence of multiple AdipoR ligands, we hypothesize that ligand diversity underlies its differential effect in T-cell immunity. To test this, we use tilapia and mouse models. Tilapia encodes AdipoR1 but lacks ADPN. Instead, an alternative adipokine, CTRP9, engages AdipoR1. We find CTRP9-AdipoR1 interaction triggers Ca^{2+} influx and activates the CaM-CaMKK β -AMPK pathway, facilitating crosstalk with TCR signaling. This cascade enhances T-cell activation, proliferation, and antimicrobial immunity by promoting glycolysis. In mice, CTRP9 similarly enhances T-cell activation, proliferation, and cytokine production and improves the efficacy of anti-CD19 CAR-T cells in eliminating B-cell lymphoma in vitro. These findings reveal an evolutionarily conserved role of CTRP9 in promoting T-cell immunity, in contrast to the inhibitory effect exerted by ADPN. Mechanistically, CTRP9 and ADPN exert distinct effects on T-cell metabolism; CTRP9 enhances T-cell glycolysis whereas ADPN suppresses it. We therefore propose ligand selectivity as determinant of AdipoR1-dependent T-cell immune outcomes.

All editorial and formatting issues were resolved by the authors.

Prof. Jialong Yang
East China Normal University
China

Dear Prof. Yang,

I am very pleased to accept your manuscript for publication in the next available issue of EMBO reports. Thank you for your contribution to our journal.

Yours sincerely,
